# *Pseudomonas aeruginosa* type IV pili actively induce mucus contraction to form biofilms in tissue-engineered human airways

Tamara Rossy[1,2], Tania Distler[1,2], Lucas A. Meirelles[1,2], Joern Pezoldt[2], Jaemin Kim[3], Lorenzo Talà[1,2], Nikolaos Bouklas[3], Bart Deplancke[2], Alexandre Persat[1,2]*

1 Global Health Institute, School of Life Sciences, Ecole Polytechnique Fédérale de Lausanne (EPFL), Lausanne, Switzerland, 2 Institute of Bioengineering, School of Life Sciences, École Polytechnique Fédérale de Lausanne (EPFL), Lausanne, Switzerland, 3 Sibley School of Mechanical and Aerospace Engineering, Cornell University, Ithaca, New York, United States of America

* alexandre.persat@epfl.ch

**Data Availability Statement:** All relevant data are within the paper and its Supporting Information files.

## Abstract

The opportunistic pathogen *Pseudomonas aeruginosa* causes antibiotic–recalcitrant pneumonia by forming biofilms in the respiratory tract. Despite extensive in vitro experimentation, how *P. aeruginosa* forms biofilms at the airway mucosa is unresolved. To investigate the process of biofilm formation in realistic conditions, we developed AirGels: 3D, optically accessible tissue–engineered human lung models that emulate the airway mucosal environment. AirGels recapitulate important factors that mediate host–pathogen interactions including mucus secretion, flow and air–liquid interface (ALI), while accommodating high–resolution live microscopy. With AirGels, we investigated the contributions of mucus to *P. aeruginosa* biofilm biogenesis in in vivo–like conditions. We found that *P. aeruginosa* forms mucus–associated biofilms within hours by contracting luminal mucus early during colonization. Mucus contractions facilitate aggregation, thereby nucleating biofilms. We show that *P. aeruginosa* actively contracts mucus using retractile filaments called type IV pili. Our results therefore suggest that, while protecting epithelia, mucus constitutes a breeding ground for biofilms.

## Introduction

Bacteria predominantly colonize their environments in the form of biofilms, dense communities of contiguous cells embedded in a self–secreted polymeric matrix [1]. The mechanisms of biofilm formation have been extensively studied on abiotic surfaces and in laboratory conditions [2,3]. In contrast, our understanding of biofilm morphogenesis in a realistic context of human infections is limited [4,5]. Biofilms from the pathogen *Pseudomonas aeruginosa* epitomize this disparity. Clinical observations show that *P. aeruginosa* forms airway–associated biofilms during acute and chronic pneumoniae in immunocompromised individuals [6,7]. Due to their clinical prevalence, *P. aeruginosa* biofilms have been meticulously investigated in vitro. Still, the architecture of in vivo *P. aeruginosa* biofilms sampled from patient sputum and lung explants is quite distinct from in vitro ones [5]. This discrepancy indicates that biofilm studies

**Funding:** This work was supported by the Swiss National Science Foundation through the Project grant number 310030_189084 (to AP) and NCCR AntiResist (to AP). The funders had no role in study design, data collection and analysis, decision to publish, or preparation of the manuscript.

**Competing interests:** The authors have declared that no competing interests exist.

**Abbreviations:** ALI, air–liquid interface; ASL, airway surface liquid; CBF, cilia beating frequency; CF, cystic fibrosis; CFU, colony–forming unit; COPD, chronic obstructive pulmonary disease; ECM, extracellular matrix; HBE, human bronchial epithelial; PFA, paraformaldehyde; scRNA–seq, single–cell RNA sequencing; SNES, Scalable Nonlinear Equations Solvers; SPIM, selective plane illumination microscope; UMAP, Uniform Manifold Approximation and Projection; WT, wild type.

in axenic environments omit critical factors of the airway mucosal surface that contribute to biofilm morphogenesis.

Epithelial tissues are lined with a hydrogel substance called mucus (Fig 1A), the first line of defense of the airway against respiratory pathogens. Dedicated goblet cells secrete gel–forming mucin glycoproteins that crosslink into a viscoelastic substance upon exocytosis to form mucus [8,9]. The mucus hydrogel mesh is impermeable to large particles, thereby functioning as a passive physical barrier [8,9]. Individuals with underlying respiratory conditions such as chronic obstructive pulmonary disease (COPD) and cystic fibrosis (CF) have aberrant mucus. At the same time, they are at risk of specifically developing chronic *P. aeruginosa* pneumonia [10]. Despite this common association, how mucus mechanics contribute to the onset and persistence of *P. aeruginosa* during infection remains unresolved.

In vitro experimentations show that mucins influence *P. aeruginosa*'s multicellular lifestyle. Mucin–coated surfaces and concentrated mucus repress *P. aeruginosa* motility [11–13], thereby favoring biofilm formation. Also, mucin polymers generate entropic forces that passively promote aggregation [14]. In other conditions, natively purified mucins can also have a negative effect on biofilms biogenesis by stimulating motility and dispersal [15–17]. These experiments each capture different chemical and physical aspects of mucus, but how all these contributions balance in vivo to influence biofilm formation has yet to be resolved. Minimally invasive experimental models that replicate physicochemical properties of the airway mucosa have the potential to bring a new perspective on this process.

Existing airway infection models have limitations that prevent mechanistic investigations of bacterial infections at the single–cell level. Tracheal explants from animal models allow dynamic studies [18–23]. They are however short–lived, displaying a rapid depletion in goblet cells after a few hours with signs of apoptosis [24]. In addition, murine and human airway mucus shows distinct composition and distribution [18]; 2D in vitro models systems based on porous membranes inserts [25–27] and lung–on–a–chip devices [28–30] are not suited for high–resolution microscopy and lack morphological accuracy. Organoids have a strong potential in recapitulating physical and biological aspects of the mucosal environment [31–33]. However, the cystic morphology of organoids prevents the establishment of an air–liquid interface (ALI) necessary to reproduce in vivo conditions. In addition, infecting organoids requires microinjection of bacterial suspension, an invasive and tedious process.

To bridge the gap between in vitro biofilm studies and clinical observations, we used a tissue–engineering approach to faithfully emulate the mucosal environment of the airway in the lab. We engineered AirGels (airway in gels): human lung epithelial tissues supported by a tubular collagen/Matrigel extracellular matrix (ECM) scaffold [34,35]. We demonstrate that AirGels recapitulate key features of the human airway epithelium, including accurate cell types, mucus secretion, and ciliary beating. We can noninvasively infect AirGels with *P. aeruginosa* while maintaining the ALI to image biofilm formation at high spatiotemporal resolution. Using this new infection model, we found that *P. aeruginosa* forms biofilms on mucus via a previously unknown mechanism. By tracking live biofilms in situ, we found that *P. aeruginosa* aggregate with one another by actively contracting mucus. Using a combination of simulations and biophysical experiments in selected mutants, we show that *P. aeruginosa* uses long and thin motorized filament called type IV pili (T4P) to generate the force necessary to contract mucus.

## Results

### AirGel: A tissue–engineered airway infection model

We grew AirGels from primary human bronchial epithelial (HBE) cells, which expand to confluence on the cylindrical cavity of the ECM scaffold (Fig 1B). An elastomeric microfluidic

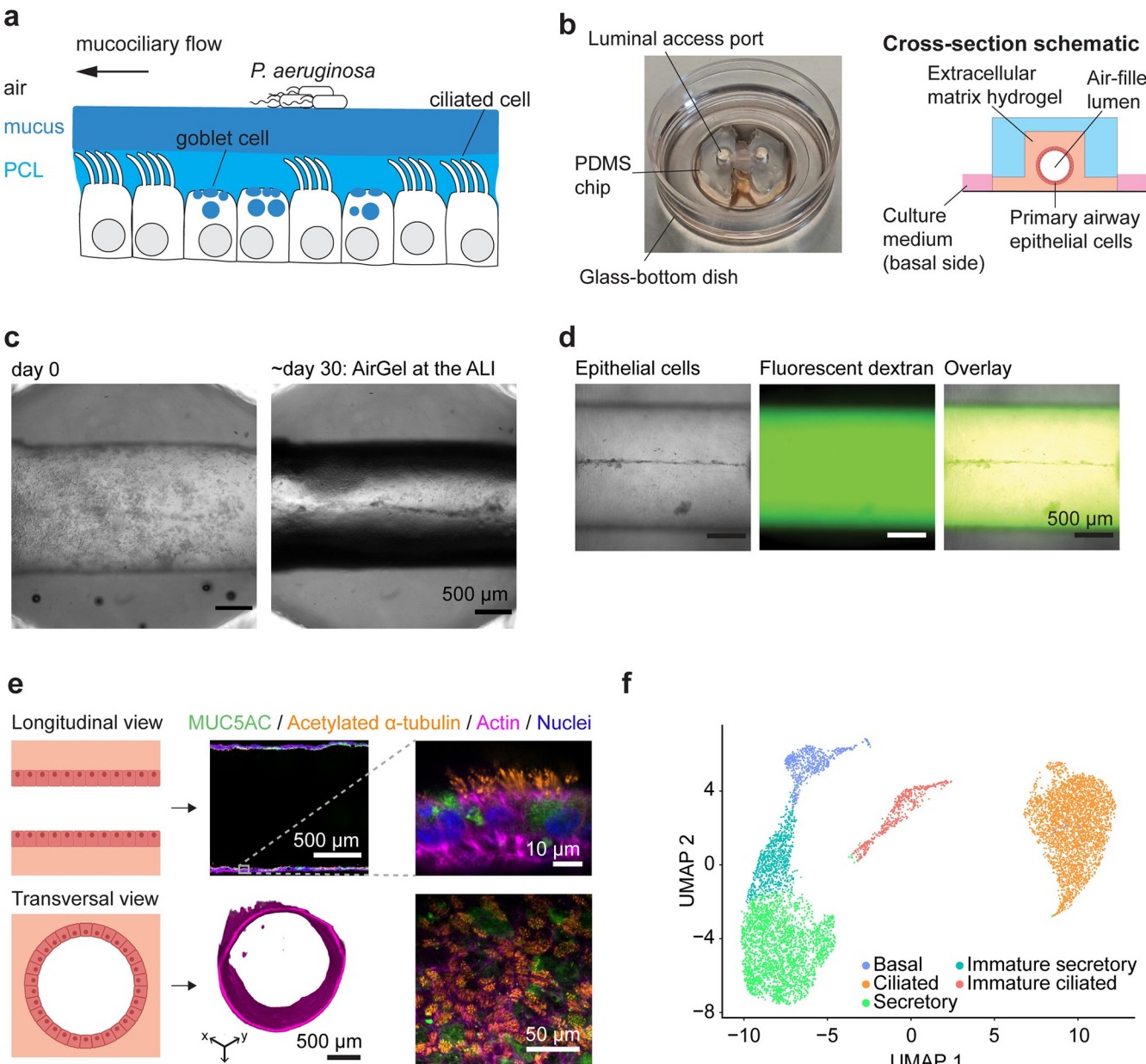

**Fig 1. A tissue–engineered airway as a novel infection model.** **(a)** Simplified representation of the airway mucosa. Mucus–secreting goblet cells and beating ciliated cells are essential to generate mucociliarly clearance, which transports inhaled pathogens out of the airway. PCL, periciliary layer. **(b)** Picture and schematic of an AirGel chip. **(c)** Brightfield image of an AirGel on the day of HBE cell were seeded (left) and at the ALI after 30 days in culture (right). **(d)** AirGels permeability measurement by dextran assay. The brightfield image (left) shows the epithelial cells lining the lumen; the epifluorescence image (center) shows signal from the fluorescent 4 kDa dextran that does not cross the epithelial barrier, as shown in the overlay picture (right). **(e)** Longitudinal cross–sectional images of immunostained differentiated AirGels. Confocal images show the gel–forming mucin MUC5AC (green) and acetylated α–tubulin labeling cilia (orange) along with the actin dye phalloidin (pink) and nuclear dye DAPI (blue). The transverse cross section 3D image was reconstituted from SPIM data for actin fluorescence. The bottom right panel is a maximal intensity projection of a z–stack acquired in the curved lumen. **(f)** scRNA–seq identifies cell type diversity of AirGels. UMAP embedding of cells pooled from 3 differentiated AirGels, subjected to scRNA–seq profiling. The data underlying this figure can be found in S1 Data. ALI, air–liquid interface; HBE, human bronchial epithelial; scRNA–seq, single–cell RNA sequencing; SPIM, selective plane illumination microscope; UMAP, Uniform Manifold Approximation and Projection.

chip maintains AirGels and allows for luminal access. The ECM geometry guides epithelial architecture, enabling morphological customization of AirGels. Here, we designed and optimized AirGels to enable high–resolution fluorescence microcopy to monitor infection dynamics at the single bacterium level in live tissue. Maintaining an ALI in the lumen promotes epithelial cell differentiation and reproduces the physiological conditions encountered in the airway. We therefore optimized the matrix formulation so that AirGels remain stable at the ALI upon removal of culture medium from the lumen, thereby biologically and physically replicating the airway environment (Fig 1C).

AirGel epithelia are tight and impermeable (Fig 1D). Single–plane illumination microscopy images show that mature AirGels form tubular epithelial tissue, recapitulating the architecture and dimensions of a human small bronchus (Fig 1E) [36,37]. We characterized HBE cell differentiation in 34–day–old AirGels. Immunofluorescence highlighted an abundant population of mucus–producing goblet cells and ciliated cells (Fig 1E). To quantify the abundance of each cell type, we performed single–cell RNA sequencing (scRNA–seq) of mature AirGels. We identified 5 main clusters (Figs 1F and S1): basal cells (8%), ciliated cells (41%), secretory cells (34%), as well as immature ciliated (7%) and immature secretory cells (which are also sometimes defined as suprabasal cells) (10%). AirGels therefore reproduce the cellular composition and histological signature of human airway epithelia [38–41] and more specifically the distal human airway [42].

Given its prominent function in host–microbe interactions, we carefully characterized the architecture of mucus in AirGels. Immunofluorescence against the airway gel–forming mucins MUC5AC and MUC5B showed the presence of extracellular mucus in the form of thick luminal filaments (Figs 2A and S2). We also observed similar fiber–like mucus architecture in live AirGels by staining with the fluorescently–labeled lectin jacalin [21]. These fibers recapitulate the mucus architecture observed in porcine and murine tracheal explants [18,20,21]. We then characterized AirGel mucociliary clearance functions. Measurements of cilia beating frequency (CBF) in AirGels were indistinguishable from previous ex vivo measurements (Fig 2B and S1 Video) [9,43–45]. In addition, AirGel cilia generated a directional flow whose clearance velocity was in the lower end of the physiological range (Figs 2C and S3) [9,18,23]. We attribute the discrepancies to different experimental conditions. For example, since AirGels are curved, our in–plane visualizations image particles that are at different heights from the epithelium surface. As a result, only a fraction of these particles lies in the appropriate range for maximal clearance. Consequently, median velocities we measured tend to underestimate of the actual clearance speed. However, we measured that maximum velocities are similar between Transwells and AirGels (S3B Fig). Humidity is another potential cause for these differences, since it is known to impact mucociliary clearance in mouse lungs [23]. Overall, though, AirGels reproduce biological, physical, and dynamic parameters of the human airway including its tube–shape, all in a system allowing for live imaging of host–pathogen interactions at high resolution.

## *P. aeruginosa* rapidly forms mucus–associated biofilms in AirGels

To visualize biofilm formation in a realistic airway mucosal context, we inoculated *P. aeruginosa* constitutively expressing the fluorescent protein mScarlet in the lumen of AirGels maintained at the ALI. After 13 h of incubation, we observed that bacteria had extensively colonized the mucosal surface. *P. aeruginosa* formed interconnected bacterial clusters colocalized with mucus within the airway surface liquid (ASL) between epithelial cells and the ALI (Fig 3A). In dynamic visualizations, bacteria remained attached to mucus despite movements induced by beating cilia (S2 Video). Since *P. aeruginosa* takes days to form biofilms in vitro, we were

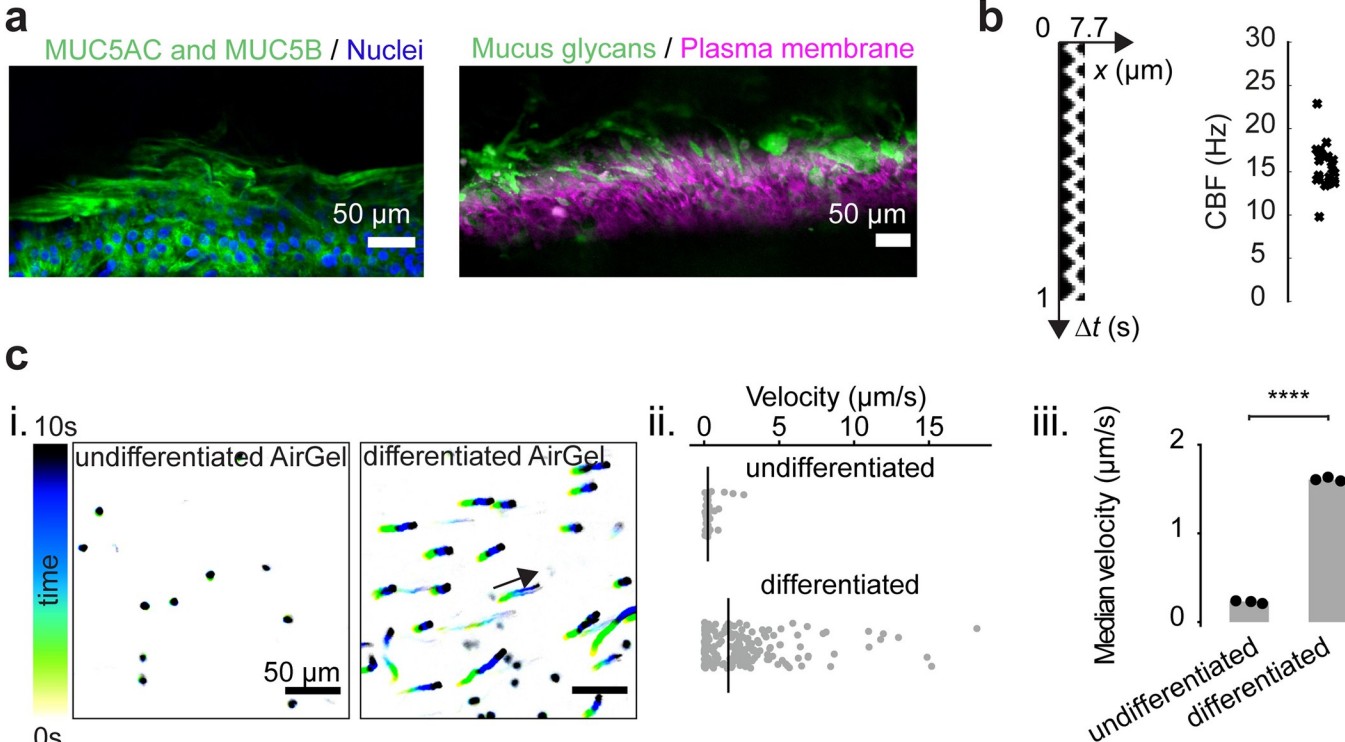

**Fig 2. Mucociliary function of AirGels.** (a) Extracellular luminal mucus in AirGels in methacarn–fixed (left) and live configurations (right). Stainings were done with antibodies against MUC5AC and MUC5B gel–forming mucins, as well as the fluorescent lectin jacalin (which targets glycans), respectively. (b) CBFs of 5 different AirGels, measured by tracking the oscillations of fluorescent beads attached to cilia. The kymograph on the left shows the trajectory of such a bead during one second. (c) Mucociliary clearance in AirGels. (i) Trajectories of fluorescent microparticles in the lumen of 1 undifferentiated and 1 differentiated AirGel. (ii) Corresponding velocity distributions. Black lines indicate the median velocity. (iii) Median particle velocities for 3 differentiated and undifferentiated AirGels show the contribution of cilia beating in clearance. Each data point corresponds to the median in each experiment; the gray bar shows the median of triplicates. Statistics: independent samples Student *t* test with Bonferroni correction ($p < 10^{-7}$). The data underlying this figure can be found in S2 Data. CBF, cilia beating frequency.

surprised to see these communities form only within a few hours in AirGels [46]. We therefore went on to investigate the mechanisms of biofilm formation on mucus.

We infected AirGels with exponential *P. aeruginosa* cultures (multiplicity of infection approximately 10) and imaged biofilm biogenesis at the single–cell level using confocal spinning disk microscopy. *P. aeruginosa* already formed aggregates a few hours after inoculation (Figs 3B and S4). While the mucus surface was initially largely devoid of bacteria, half of it was covered by *P. aeruginosa* multicellular structures after 5.5 h of infection (Fig 3C). Bacterial clusters with the same architecture also formed in the absence of jacalin staining, confirming these biofilms do not form through labeling artifacts (S5 Fig). To confirm the pivotal role of mucus in biofilm formation, we infected a non–differentiated AirGel that does not produce mucus. In the absence of a protective mucus layer, epithelial cells were more vulnerable to *P. aeruginosa* infection (S6 Fig). Bacteria breached through the epithelial barrier and invaded the underlying ECM. *P. aeruginosa* did not form 3D multicellular structures in the ASL. This further demonstrates the role of mucus hydrogel as a substrate for biofilm formation in differentiated AirGels, and at the same time highlights its protective function.

Our data suggests that *P. aeruginosa* forms biofilms in the airway by attaching to mucus at early stages of infection. To further explore the biophysical mechanisms of biofilm formation, we harvested mucus to perform ex situ visualizations. However, we could not observe the

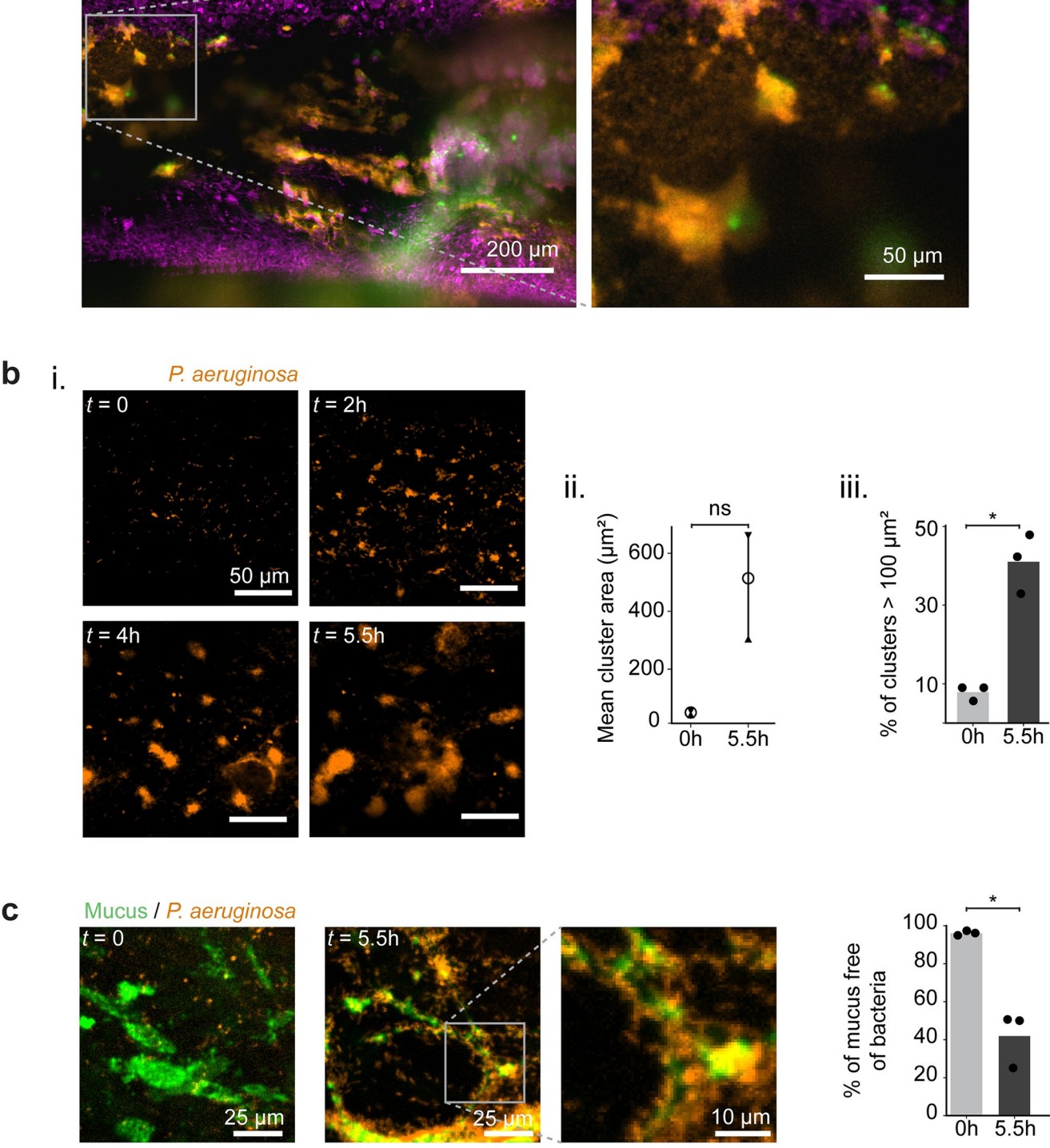

**Fig 3. *P. aeruginosa* rapidly forms mucus–associated biofilms. (a)** *P. aeruginosa* infection of a 62–day–old AirGel. Confocal images were acquired 13 h after inoculation. *P. aeruginosa* constitutively expresses the fluorescent protein mScarlet. The plasma membrane of epithelial cells was stained with CellMask Deep Red (pink). Mucus was stained with jacalin (green) shortly before infection. **(b)** (i) Maximal intensity projection images show *P. aeruginosa* biofilm formation within hours. (ii) Mean biofilms cluster area for 3 AirGels. The bar indicates the range between the maximum and minimum of the 3 means. The circle represents the mean of the means. (iii) Percentage of clusters that were larger than 100 μm² in each replicate (black dots). The bars

represent the mean across replicates. Statistics: paired samples Student $t$ test with Bonferroni correction ($p$ = 0.051 and $p$ = 0.01). **(c)** *P. aeruginosa* rapidly colonizes mucus surfaces. Images show maximal intensity projection of confocal stacks at $t$ = 0 and $t$ = 5.5 h post–inoculation. The graph quantifies the proportion of mucus not occupied by bacteria. Gray bars show the mean of triplicates. Statistics: paired samples Student $t$ test with Bonferroni correction ($p$ = 0.02). The data underlying this figure can be found in S3 Data.

formation of *P. aeruginosa* biofilms on mucus extracted from HBE cultures (S7 Fig and S3 Video). We attribute this discrepancy to perturbations in mucus mechanical integrity when extracted from the epithelium and immersed in buffer. This difference highlights the importance of investigating microbe–mucus interactions in a native mucosal context such as the one established in AirGels.

To understand how biofilms form on native mucus, we therefore inspected the different steps of their formation in AirGels. To nucleate in vitro biofilms, *P. aeruginosa* cells navigate the surface of abiotic materials using twitching motility, which promotes the formation of aggregates [47]. Fast imaging of single cells shows that *P. aeruginosa* moves with twitching–like trajectories at the surface of mucus fibers (S4 Video). Single cells aggregated into small clusters within 2 h of colonization (Fig 3B). These small multicellular clusters subsequently moved along mucus fibers to eventually fuse into larger biofilms (Fig 4A). This caused a cascade of cluster fusion events that sped up biofilm formation (Fig 4A and S5 Video). We tracked aggregate fusion in kymographs highlighting the correlation between mucus and bacterial displacements (Fig 4B). The size of individual clusters remains approximately constant during motion and fusion, showing aggregate fusion predominates over bacterial growth. After 6 h of aggregation and fusion, dense biofilms are formed.

We found that during biofilm formation, the mucus surface area tends to decrease compared to an uninfected control (Figs 4C and S8). The distances between landmarks in a mucus patch decreased over time (Fig 4D), demonstrating that mucus contracts during biofilm formation. We therefore hypothesized that mucus contraction speeds up biofilm formation by bringing *P. aeruginosa* cells closer to each other. Ultimately, these cells would become contiguous to form small aggregates. By carrying on mucus contraction, these aggregates would then fuse to each other. Since laboratory *P. aeruginosa* strains can differ substantially from strains infecting human individuals [48], we infected an AirGel with *P. aeruginosa* isolated from the bronchoalveolar lavage fluid of a CF patient. This strain colonized and contracted mucus within hours, in a manner similar to PAO1 (S9 Fig and S6 Video). To substantiate the physical contraction mechanism, we investigated how *P. aeruginosa* could restructure mucus during attachment and colonization.

### *P. aeruginosa* forms biofilms on mucus using T4P

We envisioned 2 mechanisms for bacteria–induced mucus deformations: degradation or direct mechanical contraction. *P. aeruginosa* secretes mucinases capable of degrading gel–forming mucins [49]. Enzymatic mucus degradation could release polymers that generate entropic depletion forces promoting bacterial aggregation or that generate osmotic forces compressing mucus [14,50]. To test whether mucus degradation could drive contraction, we colonized Air-Gels with a mutant in the type II secretion system locus *xcp* that is necessary for mucin utilization [49,51]. The *Δxcp* mutant however formed biofilms similar to wild type (WT), ruling out the hypothesis of polymer–induced forces driving the formation of multicellular structures (Fig 5A).

Could *P. aeruginosa* remodel mucus by directly and actively applying force on the surface? *P. aeruginosa* can generate extracellular forces using flagella and T4P, motorized filaments that also play a role during in vitro biofilm biogenesis. In addition, T4P and flagella mediate

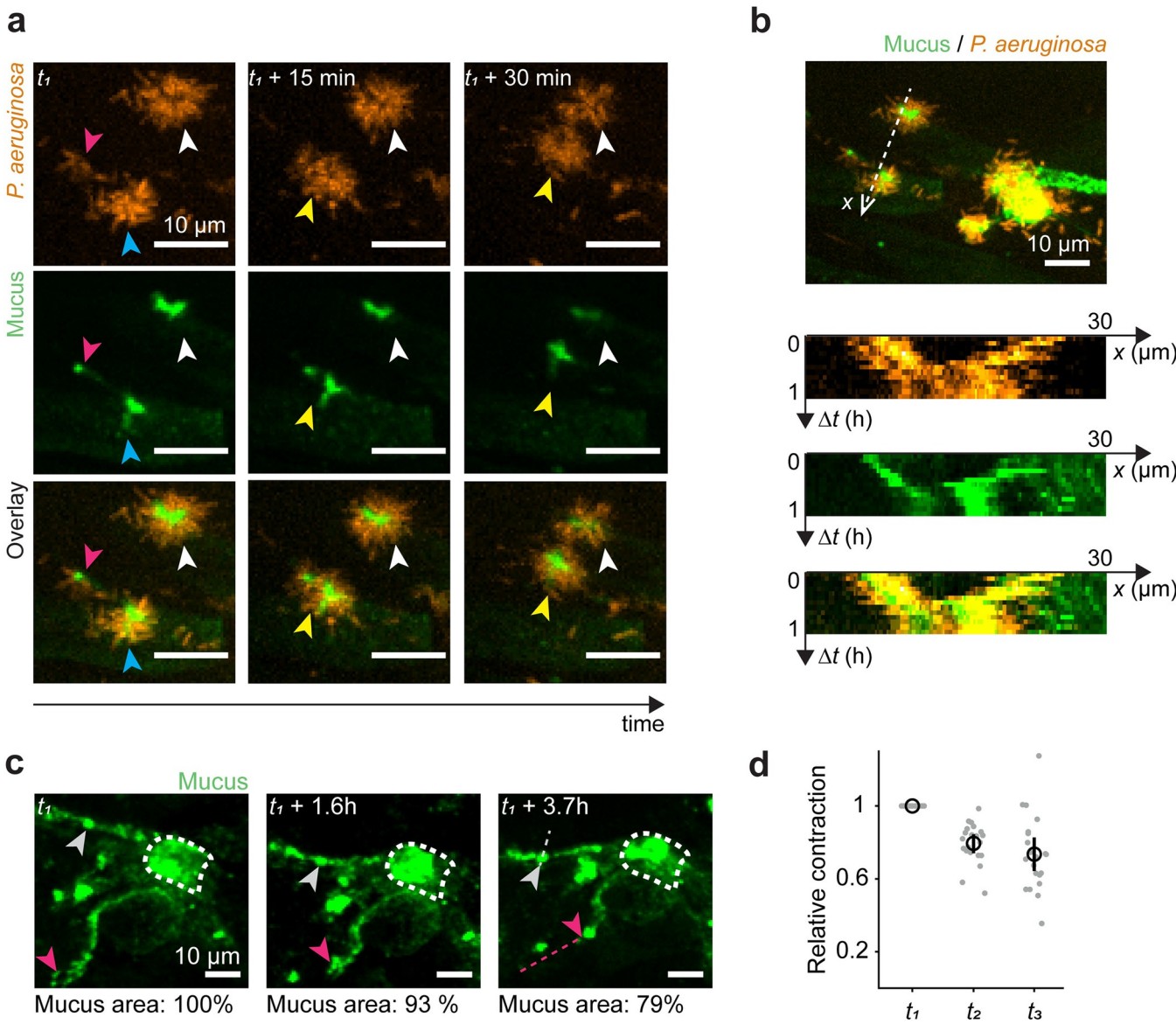

**Fig 4. Dynamics of biofilm formation on mucus. (a)** Dynamic visualization of *P. aeruginosa* cluster fusion on mucus ($t_1$ = 6.2 h). The blue and pink arrowheads show 2 aggregates that fuse within the first 15 min. The resulting cluster is indicated by a yellow arrowhead. This new cluster then moves closer to the one indicated by the white arrowhead. All images are maximal intensity projections from z–stacks. **(b)** Kymographs showing the displacement of 2 clusters along their axis of motion. The bacterial aggregate and underlying mucus traveled together at an approximate speed of 0.5 µm/min. **(c)** Time course visualization of a mucus patch in a colonized by WT *P. aeruginosa* (not displayed). Reference features and their trajectories are indicated by colored arrowheads and dashed lines ($t_1$ = 1.2 h). **(d)** Mucus contraction was quantified by tracking the distances over time between $N$ = 7 reference features in the mucus patch. The distances were normalized to the initial time point. They decrease over time, indicating contraction of the mucus patch. $t_1$, $t_2$, $t_3$ refer to the time points shown in panel c. Black circle: mean. Black line: standard deviation. The data underlying this figure can be found in S4 Data. WT, wild type.

single–cell interactions with mucins [12,15,52–54]. To investigate their functions in the context of biofilm formation on mucus, we infected AirGels with *P. aeruginosa* mutants lacking flagella (*ΔfliC*) and T4P (*ΔpilA*). The *ΔfliC* mutant formed biofilms that were indistinguishable from WT (Fig 5A). By contrast, *ΔpilA* cells did not form multicellular structures, indicating T4P play a role in mucus–associated biofilm formation. Since T4P may bind to glycans present on mucins [53,54], weaker cell attachment to mucus could cause a decrease in aggregation of *ΔpilA*. Yet, colocalization shows that the *ΔpilA* mutant is still able to attach efficiently to mucus

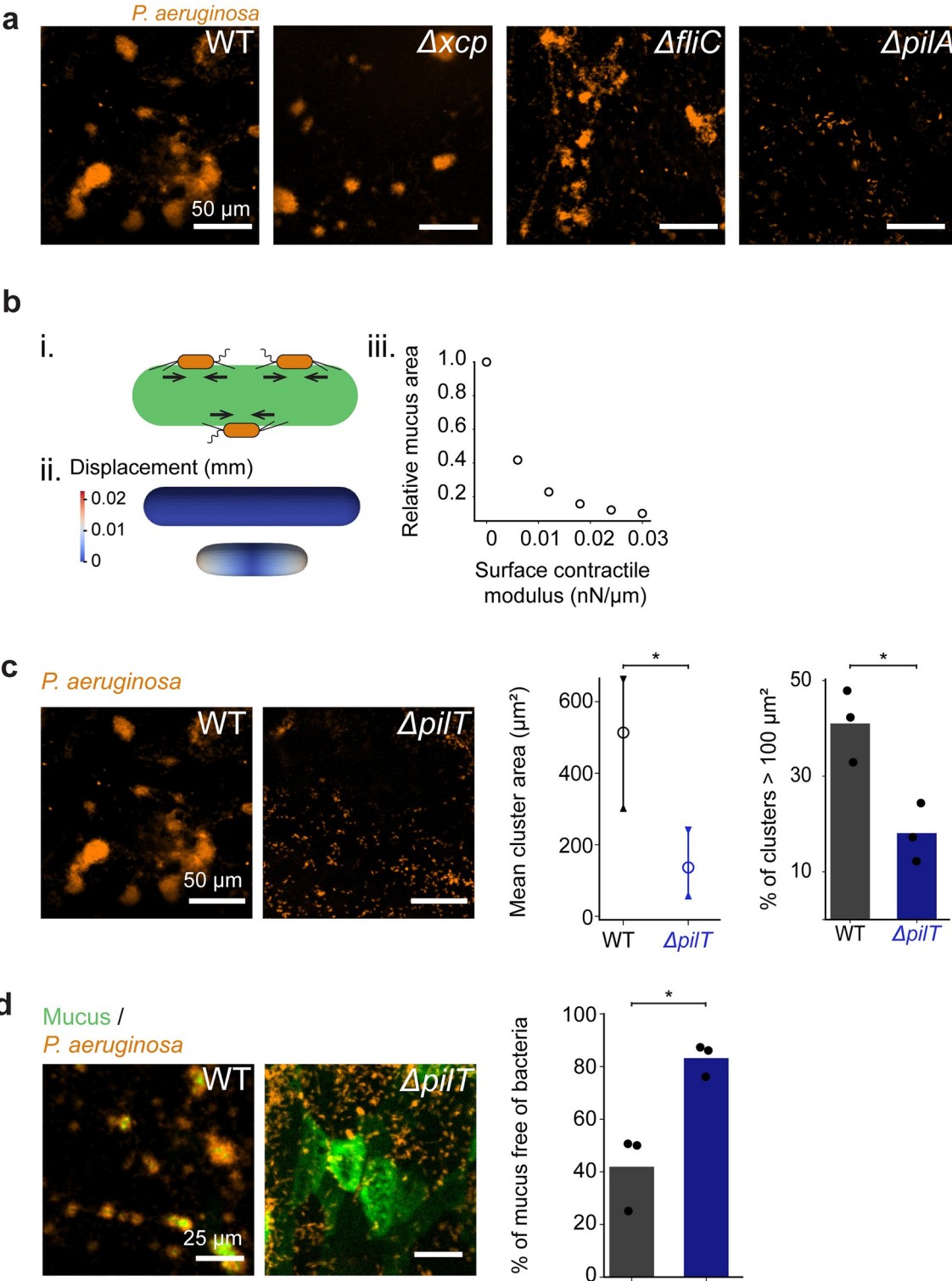

**Fig 5. Type IV pili retraction promotes mucus contraction. (a)** Biofilm formation of PAO1 mutants unable to degrade mucus or to generate force (*t* = 5.5 h). Both the *Δxcp* mutant (lacking type II secretion system necessary for secretion of mucinases) and the *ΔfliC* mutant (lacking flagella) form biofilms similar to WT. In contrast, the *ΔpilA* mutant lacking T4P was unable to form biofilms in AirGels. **(b)** Finite element simulations of mucus deformation during surface contraction. (i) Schematic representation of *P. aeruginosa* applying contractile force on mucus by retracting T4P. (ii) Finite element simulation of deformation of a mucus cylinder at

rest (top) and under active surface stress (bottom). Colormap indicate displacement of surface elements. (iii) Relative mucus area as a function of surface contractile modulus. As the surface contractile modulus increases, the relative area of mucus decreases. **(c)** T4P retraction is necessary for biofilm formation. Images compare biofilms from WT *P. aeruginosa* and from the *ΔpilT* mutant unable to retract T4P ($t$ = 5.5 h). *ΔpilT* cluster area and percentage of large clusters is significantly smaller than WT ($N$ = 3). Statistics: independent samples Student *t* test with Bonferroni correction ($p$ = 0.035 and $p$ = 0.015). **(d)** Mucus does not rearrange during *ΔpilT* colonization ($t$ = 5.5 h). Most of the mucus surface remains free of bacteria during *ΔpilT* colonization ($N$ = 3). Statistics: independent samples Student *t* test with Bonferroni correction ($p$ = 0.01). The data underlying this figure can be found in S5 Data. WT, wild type.

(S10 Fig and S7 Video). We therefore envisioned a mechanism where T4P generate retractile forces that contract mucus from the surface, ultimately speeding up *P. aeruginosa* aggregation and cluster fusion.

To physically explore this scenario, we ran nonlinear finite element simulations wherein mucus is treated as a hyperelastic material [55]. The mechanical action of *P. aeruginosa* T4P at the mucus surface is considered through the introduction of an active surface stress. The simulations recapitulated the experimental observations of mucus hydrogel contraction during *P. aeruginosa* colonization (Fig 5B). Simulations also predict that the steady–state mucus area decreases with the magnitude of the surface contractile modulus. This suggests that the more T4P retract, the more *P. aeruginosa* compresses mucus. To experimentally validate this model, we visualized AirGels colonization by a *ΔpilT* mutant that produces T4P that cannot retract, mimicking conditions of zero contractile modulus. *P. aeruginosa ΔpilT* could still associate with mucus and form a few small clusters, but clearly failed to form mucus–associated biofilms to the same extent as WT (Fig 5C and 5D), which was coherent with simulations. These results show that T4P retraction is necessary for biofilm formation on mucus and is consistent with a scenario where retraction compresses the mucus substrate.

To further support the surface contraction model, we tested the prediction that deformations increase further with surface contractility. We imaged AirGel colonization by the hyper-piliated *P. aeruginosa* mutant *ΔpilH*, whose T4P retraction frequency is approximately twice the one of WT (S11 Fig). *ΔpilH* formed biofilms more rapidly than WT: we observed dense aggregates as early as 2 h, while we only did after 4 h for WT (Fig 6A and 6B). In addition, *ΔpilH* induced more rapid mucus contraction than WT (Fig 6C and S8 and S9 Videos), consistent with simulations. After 5.5 h, WT and *ΔpilH* biofilms had similar morphologies and size, suggesting biofilm fusion reaches a physical limit most likely due to packing at the mucus surface. To control that the observed differences did not arise from growth rate variation between the strains, we quantified colony–forming units (CFUs) of WT, *ΔpilT*, and *ΔpilH* before infection and after 5 h of growth at the airway mucosa (S3 Table). All strains grew at indistinguishable rates.

Overall, our results support a model where *P. aeruginosa* contracts the surface of mucus by active T4P retraction. Single cells initially twitch on mucus to form small aggregates. The static aggregate collectives generate forces from T4P that are sufficient to deform their substrate, driving large–scale mucus contraction. By contracting, mucus brings aggregates closer. They eventually fuse and form biofilms (Fig 7).

## Discussion

Most investigations of host–pathogen interactions have so far mainly relied on animal models and immortalized cell lines. Their limitations have been an obstacle to establish a holistic understanding of infections. By developing AirGels, we provide the community with a 3D airway infection model that expresses relevant cell types, secretes mucus, and is compatible with high–resolution imaging in presence of an ALI. As a result, AirGels have a strong potential in bridging the gap between in vivo and in vitro investigations of airway infections. For example,

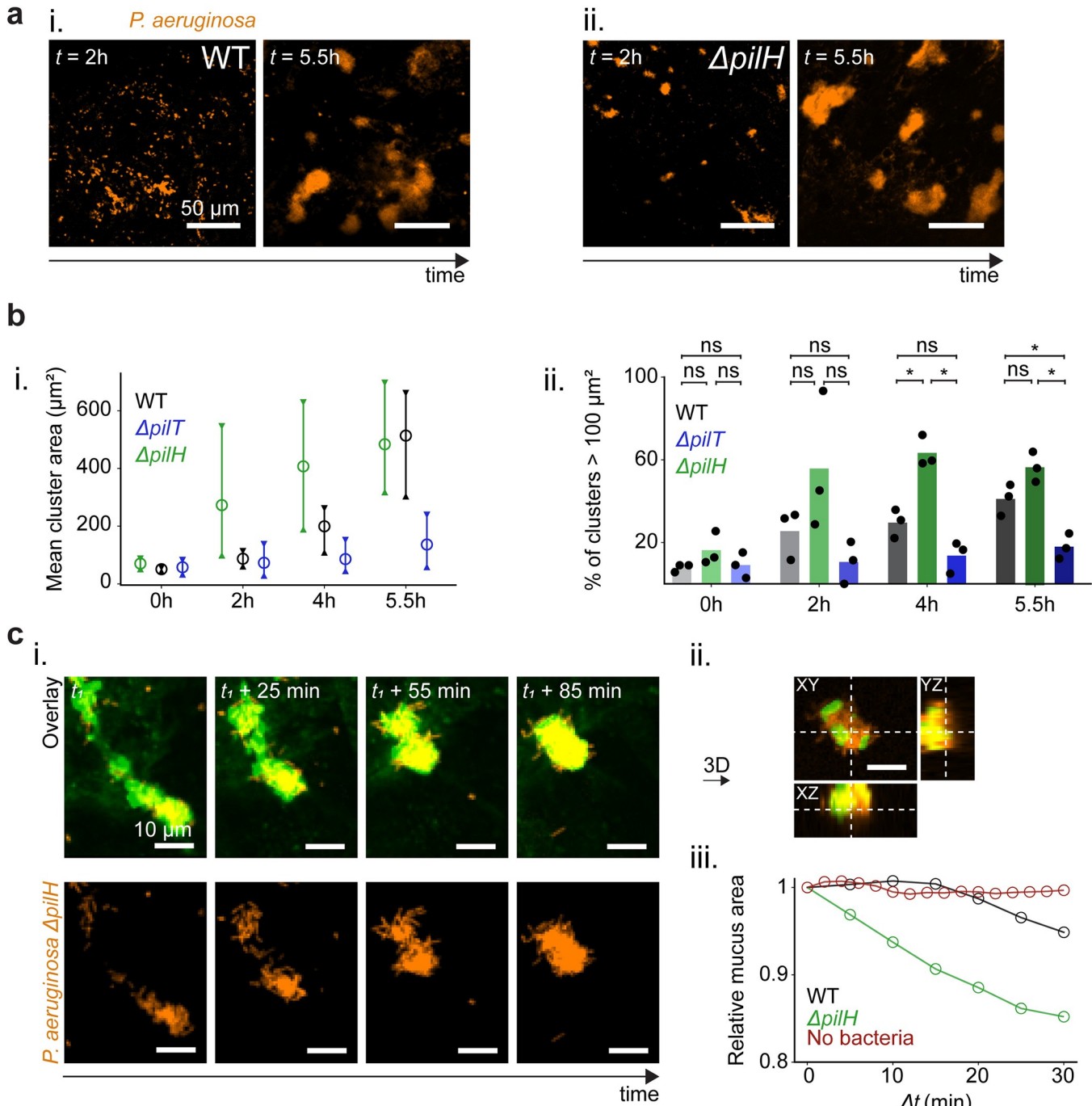

**Fig 6. Increased T4P retraction speeds up biofilm formation and mucus contraction. (a)** Increased T4P activity speeds up biofilm formation on mucus. Comparison of biofilm formation by the *ΔpilH* mutant with hyperactive T4P vs. WT, at *t* = 2 h and *t* = 5.5 h after inoculation. *ΔpilH* already forms small biofilms after 2 h. **(b)** (i) Kinetics of biofilm size for WT, *ΔpilT*, and *ΔpilH*. For each strain, we infected 3 AirGels from a healthy donor. Bars represent the range between the maximum and minimum of the means from triplicates, circles represent the overall mean. (ii) Comparison of percentage of large clusters for WT, *ΔpilT*, and *ΔpilH*, over time. Statistics: one–way ANOVA for each time point, followed by a post hoc Tukey test if the null hypothesis was rejected. At *t* = 4 h, the differences between WT and *ΔpilH* (*p* = 0.003) and between *ΔpilH* and *ΔpilT* (*p* = 0.001) were significant. At *t* = 5.5 h, the differences between WT and *ΔpilT* (*p* = 0.02) and between *ΔpilH* and *ΔpilT* (*p* = 0.001) were significant. **(c)** *ΔpilH* dramatically contracts mucus. (i) Timelapse images showing an event of mucus contraction by *ΔpilH*. (ii) Orthogonal views of the bacteria–covered mucus cluster at *t₁* + 85 min, showing that PAO1 *ΔpilH* cells surround mucus. (iii) Relative mucus area changes measured during 30 min for WT and *ΔpilH*; since *ΔpilH* starts aggregating and remodeling mucus earlier than WT, the starting points of the recording differed (*ΔpilH*: 2.5 h, WT: 6.2 h, negative control: 8.1 h). Images are maximal intensity projections of z–stacks throughout the figure except for the orthogonal projection in H. The data underlying this figure can be found in S6 Data. WT, wild type.

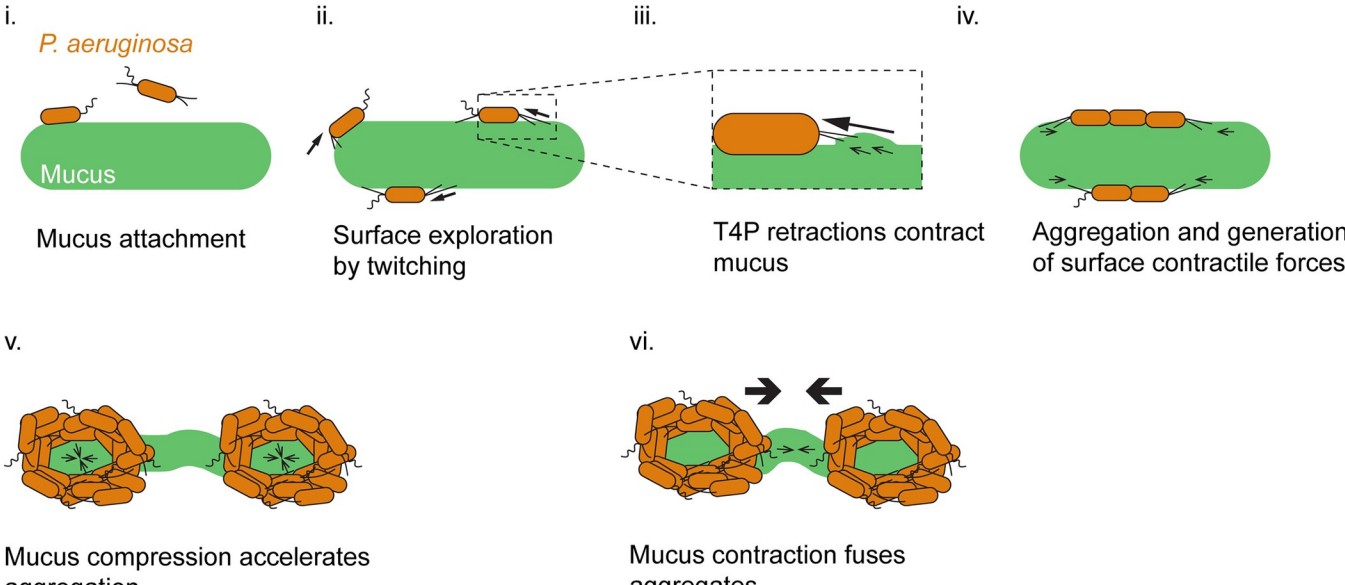

**Fig 7. Proposed model for the formation of mucus–associated biofilms by *P. aeruginosa*.** (i) Single bacterial cells attach to mucus. (ii) *P. aeruginosa* navigates the mucus surface using T4P–dependent twitching motility. (iii) T4P retraction locally contracts mucus. (iv) Surface exploration promotes encounters between single cells. This initiates aggregation and the formation of small clusters. These clusters remain static on mucus. As single cells in aggregates use T4P to pull on and contract mucus, they generate surface contractile forces. (v) Collective retraction of T4P from many cells compacts bacterial aggregates. (vi) Still under the action of retractile T4P forces, aggregates further contract mucus to initiate fusion into biofilms.

since AirGels are modular, we envision that engineering refinements could improve their suitability as an infection model for a wide range of organisms and incorporation of immune cells. AirGel benchmarking also allowed us to identify how important physiological parameters such as mucociliary clearance are subject to variability and strong dependencies on external factors. AirGels thus provide a useful new tool to perform comprehensive and mechanistic studies of the dependence of airway physiology on factors such as temperature, humidity, drugs, and chemical stressors.

By leveraging AirGels, we found that *P. aeruginosa* forms biofilms via an active mechanism of mucus remodeling. *P. aeruginosa* attaches to mucus and subsequently uses T4P to generate surface contractile forces. As a result, the mucus gel contracts, effectively reducing its area and bringing mucus–bound bacteria closer to each other. Although the classical view of airway infections associates biofilms with chronic infections and planktonic cells with acute infections, recent work has demonstrated the coexistence of these bacterial lifestyles in sputum samples from both acutely and chronically infected patients [7]. Our observations of early biofilm formation in AirGels from planktonic cells is therefore consistent with these clinical observations. However, in vitro, the effect of mucus on biofilm formation depends on the experimental model: while native mucins in solution inhibit biofilm formation [15–17], studies with full mucus or commercial mucins instead report increased aggregation [11,12,14]. This demonstrates that mucus–pathogen interactions vary dramatically depending on the model system used, thereby highlighting the importance of carefully reproducing and controlling relevant parameters in vitro.

During *P. aeruginosa* infections, the planktonic population tends to have stronger cytotoxicity towards the host compared to the biofilm population [6]. Therefore, improving biofilm formation would reduce the pathogenic power of *P. aeruginosa* populations. Forming biofilms early on could however provide other fitness advantages to *P. aeruginosa* in the non–hospitable

airway environment. For example, bacterial aggregation could reduce *P. aeruginosa*'s suscepti-bility to neutrophils that are rapidly recruited to the mucosal surface during infection [56,57]. At the same time, forming biofilms increases *P. aeruginosa* tolerance to antibiotic treatment and promotes the emergence of resistant mutants [58]. There is however an upside for the host: mucus adsorbs a large proportion of the planktonic *P. aeruginosa*, keeping them away from the epithelium. Our results therefore highlight the duality of mucus: protecting the air-way epithelium from acute infections, while providing a fertile ground for biofilm formation that favors chronic infections.

T4P play multiple functions during biofilm formation in many species. In *P. aeruginosa*, successive T4P extension and retraction power twitching motility on surfaces [59,60]. This allows freshly attached cells to explore the environment, stimulating cell–cell encounters that nucleate aggregation [60]. These microcolonies ultimately mature into full biofilms. This model however falls short on soft surfaces. Hydrogels with low stiffness limit the transmission of T4P traction force to the surface thereby impairing twitching motility, but at the same time still enable biofilm formation [61]. Mucus contractions induced by *P. aeruginosa* show that T4P–generated forces can remodel soft materials as well. Our model presents similarities with the mechanisms by which *Neisseria meningitis* forms and maintains microcolonies. *N. menin-gitis* use T4P to maintain biofilm cohesion, where single cells connect to one another via pili–pili interactions. Their retraction is critical to the cohesion of the community [62,63]. In that sense, T4P retraction promoted the emergence of contractile forces at the scale of the multicel-lular structure, thereby improving its cohesion. In addition to highlighting a new mode of bio-film formation, this mechanism provides additional evidence that bacteria can mechanically remodel the host microenvironment [64].

By emulating the mechanical environment experienced by airway pathogens during muco-sal colonization, we have identified a new mechanism by which bacteria form biofilms. This demonstrated that alternative approaches that leverage engineered microenvironments will help us better comprehend bacterial physiology in realistic infection contexts. This could ulti-mately allow the discovery of overlooked aspects of infections. In addition, it has the potential to provide the community with new tools and knowledge to develop novel therapeutic strate-gies against antibiotic–resistant infections.

## Materials and methods

### Ethics statement

No consent from the ethics committee was required for the clinical specimen as only anon-ymized health–related data was used.

### AirGel chip fabrication

**3D printed mold.**   The mold for the PDMS chips was designed in Autodesk Inventor Pro-fessional 2021. This mold was then 3D printed by Multi Jet Modeling on a Connex 500 printer (Objet) using VeroClear resin at the Additive Manufacturing Workshop (AFA) at EPFL. In order to remove uncured resin that could interfere with subsequent PDMS polymerization, we treated our mold by soaking it in deionized water for 2 h; then, we incubated it for 18 h in an oven set to 85˚C; finally, we washed it with deionized water and dishwashing soap before let-ting it dry.

**PDMS chip.**   PDMS (Sylgard 184, Dow Corning) was casted on the mold and cured at 60˚C for about 1 h and 30 min. We then used a scalpel to cut out each chip individually and carefully remove it from the mold. In parallel, we prepared PDMS rods to pattern the lumens according to a published protocol [34]. In short, we filled gauge 14 needles (Sterican 2.1 × 80

mm, B. Braun) with PDMS and cured it as described above. We then used pliers to break the needles and extract the PDMS rods; their diameter was approximately 1.6 mm (i.e., inner diameter of the needle). We used a scalpel to cut them into 8–mm long pieces. Then, PDMS chips and rods were briefly immersed in isopropanol, left to dry, and cleaned using tape. Afterwards, the rods were inserted into the chips using tweezers; the assembled devices were subsequently autoclaved. They were then plasma bonded to either glass–bottom dishes or glass–bottom 6–well plates (1.5 coverslip, glass diameter 20 mm, MatTek) in a ZEPTO plasma cleaner (Diener electronic). Note that the chips contained thin PDMS membranes at the bottom of their inlet reservoirs (obtained owing to a shallow cavity in the 3D printed mold), so that the rod was not in direct contact with the underlying coverslip. Finally, we exposed the chips to 2 cycles of UV sterilization in a biosafety cabinet.

**Extracellular matrix.** All the following steps were performed in a biosafety cabinet to maintain sterility. We treated the chips inner surfaces to promote adhesion of the gel following a published method [35]. This consisted in a 10 min exposure to 2% polyethyleneimine (Sigma–Aldrich) followed by a 30 min treatment with 0.4% glutaraldehyde (Electron Microscopy Sciences) for 30 min. The chips were then rinsed once with Milli–Q water (Merck Millipore). Afterwards, the ECM hydrogel was prepared on ice. We first neutralized high–density rat tail type I collagen (approximately 10 mg/ml, Corning) to a final concentration of 8 mg/ml. To do this, we mixed 30 µl 10× PBS, 5.4 µl NaOH 1 M, 29.6 µl Milli–Q water, and 235 µl collagen with a positive displacement pipette (Gilson). The neutralized collagen was then mixed with high–concentration growth factor reduced Matrigel (approximately 21 mg/ml, Corning) in a 75:25 ratio (100 µl Matrigel for 300 µl of neutralized collagen). The resulting gel was loaded into each chip from the basal access ports and placed in a humidified cell culture incubator (set to 37°C, 5% $CO_2$) during 20 min in order for polymerization to occur. Then, we pulled the rods out of the chips with tweezers, thereby shaping the lumen [34]. The final step consisted in chemically crosslinking the collagen to strengthen it. We followed a published protocol [65]: we first prepared a 0.6 M solution of N–(3–dimethylaminopropyl)–N′–ethylcarbodiimide hydrochloride (EDC, Life Technologies) and a 0.15 M solution of N–hydroxysuccinimide (NHS, Sigma–Aldrich). Then, we mixed those solutions in a 1:1 ratio and loaded 25 µl into each lumen. We left them at room temperature for 5 min before aspirating the crosslinking reagents. We then soaked the chips in Milli–Q water (apical and basal sides) overnight, at room temperature. Finally, we replaced Milli–Q water with PneumaCult–Ex Plus medium (Stemcell Technologies) at least 1 day before loading any cells in the chips, and stored the chips in a cell culture incubator.

## Cell culture

**Expansion in flasks.** We obtained primary HBE cells from Lonza (CC–2540S or CC–2540 for healthy donors, 00196979 for the CF donor). They were cultured in T–25 flasks using PneumaCult–Ex Plus medium (Stemcell Technologies) for no more than 3 passages. When they reached confluence, the cells were detached from the flask using the Animal Component–Free Cell Dissociation Kit (Stemcell Technologies) and centrifuged before being resuspended in PneumaCult–Ex Plus to a density of approximately 20,000 cells/µl.

**Loading into AirGels.** AirGels were emptied of all culture medium on the apical and basal sides. Approximately 10 to 12 µl of HBE cell suspension was loaded in the lumen of each AirGel. The chips were then placed in a cell culture incubator for 25 min, flipped upside down and incubated again for 25 min, and then finally for 15 min on each side in order to allow uniform adhesion of cells along the luminal surface. Afterwards, PneumaCult–Ex Plus was added to the lumen and on the basal side of the chips.

**Long–term culture in AirGels.**   HBE cells were expanded in AirGels with PneumaCult–Ex Plus until confluence was reached (typically 1 to 3 days). After that, apical and basal expansion medium was replaced with PneumaCult Airway Organoid Differentiation Medium (Stemcell Technologies). In order to prevent gel degradation by the HBE cells, we supplemented the medium with 5 μm of protease inhibitor GM6001 (InSolution GM6001, Merck). One day after the first addition of differentiation medium, all fluid was manually aspirated from the lumen, thereby generating an ALI. This critical step was facilitated by the aforementioned collagen crosslinking and by the large lumen diameter, which lowered capillary forces drawing medium back in the channel. AirGels could be kept in these conditions for at least 1 month; medium was replaced on the basal side every second day. Before weekends, the lumen was also filled with PneumaCult Airway Organoid Differentiation Medium, but ALI was restored every Monday morning and passively maintained for the whole week.

**Cell culture on Transwell membranes.**   For isolated mucus experiment and twitching motility imaging, we grew NHBE cells on 0.4–μm pore polyester Transwell membranes (Corning) instead of AirGels. The expansion phase and cell dissociation process were as described above. NHBE cells were loaded on Transwells at a density of approximately five·$10^4$ cells per well. When they reached confluence, they were transitioned to ALI culture conditions, i.e., with PneumaCult–ALI medium (Stemcell Technologies) on the basal side and air on the apical side.

## Live staining

To label mucus in live AirGels, we used jacalin conjugated to fluorescein (Vector Laboratories). We prepared a 50 μg/ml solution and loaded it in the lumen. We stored the chips for 30 min in a cell culture incubator before aspirating all fluid from the lumen. In addition, in order to assess epithelial integrity, we loaded a 4 kDa fluorescent dextran solution in the lumen of an 11–day–old chip and incubated it for 30 min. We then verified that all signal was localized in the lumen of the chip.

## Immunofluorescence

All steps were performed at room temperature. First, differentiated cells in AirGels were fixed with either 4% paraformaldehyde (PFA, Electron Microscopy Sciences) or methacarn, when we wanted to better preserve extracellular mucus. Methacarn was made fresh before every use as follows: 1 part glacial acetic acid (Sigma–Aldrich), 3 parts chloroform (PanReac AppliChem), 6 parts anhydrous methanol (Sigma–Aldrich). Regardless of the chemical used, the fixation step lasted for 15 min. PFA–fixed cells were then permeabilized with a 0.2% Triton X–100 solution (VWR Life Science) for 20 min. Then, we exposed all cells (i.e., PFA–and methacarn–fixed) to a blocking solution consisting of 1% bovine serum albumin (Sigma–Aldrich) during 45 min. Afterwards, we added solutions of primary antibodies to each AirGel and incubated them for 1 h. In case of PFA–fixed cells, we used rabbit anti–MUC5AC (1:100, Abcam) and mouse anti–acetylated alpha tubulin (1:250, Sigma–Aldrich); for methacarn–fixed cells shown in Fig 2A, we used the same anti–MUC5AC, together with rabbit anti–MUC5B (1:100, Sigma–Aldrich); for methacarn–fixed cells shown in S2 Fig, we replaced the rabbit anti–MUC5AC antibody with mouse anti–MUC5AC (1:100, Sigma–Aldrich). After incubation with the primary antibodies, we labeled the samples with secondary antibodies during 1 h in the dark. More specifically, we used goat anti–rabbit IgG H&L Alexa Fluor 488 (1:200, Abcam) and goat anti–mouse IgG H&L Alexa Fluor 594 (1:200, Thermo Fisher). Finally, nuclei were counterstained for 10 min with DAPI (1:1,000, Sigma–Aldrich); in addition, in PFA–fixed cells, actin was stained with Phalloidin Atto 655 (1:40, Sigma–Aldrich) for 10 min.

## Sample preparation before lightsheet imaging

To perform lightsheet microscopy on AirGels, we needed to extract the ECM gel and cells from the PDMS chip. After fixation and staining, we filled in the lumen with a 1% low–melt agarose solution in order to ensure structural integrity of the airway. We let it solidify; using tweezers, we could then carefully detach the PDMS from the glass; indeed, since the surface of the 3D printed mold was not perfectly smooth, plasma bonding was not irreversible, which we could leverage for ECM extraction. We then used a scalpel and a spatula to release the piece of ECM from the PDMS chip and later embedded it in 1% low–melt agarose. While the agarose was still liquid, we aspirated the whole gel into a 1–ml syringe (Omnifix–F, B. Braun), whose tip had previously been cut out. After the agarose solidified, we could then use the plunger to freely push the fixed AirGel in and out of the syringe, in order to image it with selective plane illumination microscope (SPIM).

## Microscopy

To image AirGels at low magnification (Fig 1C and 1D), we used a Nikon TiE epifluorescence microscope equipped with a Hamamatsu ORCA Flash 4 camera and either a 10× objective with N.A. of 0.25 or a 4× objective with N.A. of 0.1. For full channel cross–sectional imaging (Fig 1E), we used a Zeiss Lightsheet Z1 dual–sided SPIM. It was equipped with PCO Edge 5.5 cameras and a 5× magnification objective with N.A. of 0.16. All the other visualizations were acquired with a Nikon Eclipse Ti2–E inverted microscope coupled with a Yokogawa CSU W2 confocal spinning disk unit and equipped with a Prime 95B sCMOS camera (Photometrics). We either used a 20× water immersion objective with N.A. of 0.95 or a 40× water immersion objective with N.A. of 1.15. We used Imaris (Bitplane) for 3D rendering of lightsheet z–stack pictures and Fiji for the display of all the other images [66].

## Single–cell RNA–seq

**Sample processing and sequencing.** Three AirGels differentiated for 35 days were pooled to perform scRNA–seq. The AirGels were washed 3 times with PBS from the apical and basal sides before carefully detaching the PDMS chip from the dish. Epithelia were removed together with their ECM from the chip using forceps and placed in dissociation buffer (300 μl Protease from Bacillus Licheniformis (100 mg/ml, Sigma), 3 μl DNase I (10 mg/ml, Roche), 30 μl EDTA (0.5 M, Sigma), 30 μl EGTA (0.5 M, BioWorld), 237 μl sterile PBS, and 900 μl Accumax (Brunschwig)). Incubation was performed for 45 min at 37˚C except for centrifugation and pipetting steps which were performed at room temperature. Initially, the piece of gel and attached cells was disrupted by pipetting up and down 50 times every 5 min with a 200 μl filtered pipette tip. After the first 20 min of incubation, the cells were mostly detached from the gel and the cell suspension was centrifuged for 5 min at 400× g, after which the supernatant was removed. The residual volume (approximately 20 μl) was pipetted up and down 50 times every 5 min to disrupt cellular aggregates, this time with a 10 μl filtered pipette tip. Finally, the cell suspension of all 3 chips was combined and topped up to 1 ml with pre–cooled 10% BSA (Sigma–Aldrich) in PBS. From this point, all steps were performed on ice or at 4˚C. The cells were centrifuged for 10 min at 400× g. The supernatant was carefully removed and the cells were resuspended in 0.04% molecular grade BSA in PBS and filtered through a 40 μm Flowmi cell strainer (Bel–Art). The cell suspension was centrifuged once more at 400× g for 10 min. The supernatant was removed and the cells were resuspended in 50 μl 0.04% molecular grade BSA in PBS. The cell count was determined and the cells were immediately taken to the sequencing facility at EPFL (GECF).

HBE cells were then washed once in PBS 10% BSA and then once in PBS 0.04% BSA. After filtration through a 40 μm Flowmi strainer, cells were resuspended in PBS 0.04% BSA, checked for absence of significant doublets or aggregates, and loaded into a Chromium Single Cell Controller (10× Genomics) in a chip together with beads, master mix reagents (containing RT enzyme and poly–dt RT primers), and oil to generate single–cell–containing droplets. Single–cell Gene Expression libraries were then prepared using Chromium Single Cell 3' Library & Gel Bead Kit v3.1 (PN–1000268) following the manufacturer's instruction (protocol CG000315 Rev C). Quality control was performed with a TapeStation 4200 (Agilent) and QuBit dsDNA high sensitivity assay (Thermo) following manufacturer's instructions. With this procedure, the cDNAs from distinct droplets harbor a distinct and unique 10× "cell barcode."

Sequencing libraries were processed using an Illumina HiSeq 4000 paired–end Flow Cell and sequenced using read lengths of 28 nt for read1 and 91 nt for read2, at a depth of ca 60k reads/cell.

**scRNA–seq analysis.** The Cell Ranger Single Cell Software Suite v6.1.1 was used to perform sample demultiplexing, barcode processing, and 3' gene counting using 10× Genomics custom annotation of human genome assembly GRCh38 [67]. Count matrices were further processed with Seurat (version 4.1.0) [68]. All cells with less than 1,000 detected genes per cell were filtered out. Moreover, cells with more than 25% reads mapping to mitochondrial genes were removed yielding 8,651 cells passing QC. After filtering, data were default normalized and the 2,000 most variable genes identified. The expression levels of these genes were scaled before performing PCA. The following covariates were regressed out: number of UMIs and percent of mitochondrial reads Uniform Manifold Approximation and Projection (UMAP) dimensionality reduction was performed using the first 25 dimensions of the PCA and resolution set to 0.175. Cell subsets were identified based on transcriptional signatures previously identified by Plasschaert, Žilionis and colleagues [40]. One subset was comprised of cells with a shared signature between "Ciliated" and "Secretory" cells, with a total of 640 cells, indicative of doublets and were thus removed. The remainder cells, numbering 8,011 were re–embedded as described above (resolution = 0.15). GO analysis was performed for differentially up–regulated genes per cluster using *TopGO* [69].

## Quantification of cilia beating frequency (CBF)

We filled the lumens of AirGels with a 1:500 solution of yellow–green carboxylated fluorescent beads with a 2 μm diameter (FluoSpheres, Life Technologies). We incubated them for 1 h; in this time interval, despite the flow generated by ciliary beating, some beads were able to settle down and attach to cilia. We then removed all fluid from the lumen and brought the chips to the spinning disk confocal microscope. We then selected small regions of interest (50 × 50 pixels, i.e., 27.5 μm × 27.5 μm) around individual beads and recorded videos at 100 frames per second. Using Matlab R2016b (Mathworks), we computed the mean intensity of each frame over time. We computed the fast Fourier transform of each mean intensity signal, which we then used to obtain single–sided power spectra. We only kept frequencies between 1 and 30 Hz, thereby getting rid of artifacts. We finally looked for the frequency with maximal amplitude in the power spectrum, which corresponded to the CBF.

## Quantification of mucociliary clearance

Like for CBF quantification, we loaded a 1:500 solution of 2 μm FluoSpheres in the lumen of AirGels (or at the surface of Transwells, for S3 Fig). We immediately visualized them with the spinning disk confocal microscope. We recorded 10 s videos at a rate of 10 frames per second.

Then, we tracked the trajectory of individual beads with the Fiji plugin TrackMate [70,71], using the built–in simple LAP tracker. We wrote a script in a Jupyter Notebook to compute the velocity (track displacement over track duration) of each particle [72].

## Bacterial strains, plasmids, and culture conditions

We used *P. aeruginosa* PAO1 (WT or mutants, listed in S1 Table) for all the infection experiments. To obtain the clinical strain AP1889, bronchoalveolar lavage fluid of a CF patient was sent to the routine lab of the Institute of Medical Microbiology in Zurich, Switzerland. The clinical sample was cultured on different media types and *P. aeruginosa* was isolated from a selective Mac Conkey agar (Biomérieux, France) after incubation for 18 h at 37°C. Species identification was performed by MALDI–TOF MS biotyper (Bruker Daltonics, Germany) as described elsewhere [73]. No consent from the ethics committee was required for the clinical specimen as only anonymized health–related data was used. Most strains were made to constitutively express the fluorescent protein mScarlet following a published protocol using the plasmids listed in S2 Table [74]. The backbone plasmid pUC18t–Mini Tn7 with gentamycin resistance was purchased from Addgene and isolated from *E. coli* XL10 Gold by GeneJET Plasmid Miniprep Kit (Thermo Fisher). The isolated plasmid was digested with the restriction enzymes HindIII and BamHI. The Ptet promoter was amplified by PCR using *P. aeruginosa* PAO1 genomic DNA and the mScarlet gene was amplified from a preexisting plasmid. The Ptet promoter and mScarlet was then fused via Fusion PCR by overlapping extension. The resulting extended product was digested with HindIII and BamHI, and then ligated to the digested pUC18t–MiniTn7 Gm backbone. Since this plasmid included a gentamycin resistance cassette, we grew the fluorescent PAO1 strains overnight in LB medium with 30 µg/ml gentamycin. The next morning, we diluted the stationary cultures 1:1,000 in plain LB and let them grow 3 to 4 h before infecting AirGels.

## Infection of AirGels

The night before infection, AirGels were stained with the plasma membrane dye CellMask Deep Red (Life Technologies). Aside from the infection assay shown in Fig 3A and S2 Video, which was performed in a CF AirGels, all infections were run with cells from healthy donors. The dye was diluted to 5 µg/ml and loaded in both the apical and basal compartments. The next morning, the lumen was again exposed to air for 3 to 4 h. Mucus was stained with jacalin as described above, and all luminal fluid was then aspirated. Finally, we infected AirGels with mScarlet *P. aeruginosa*. We measured the optical density of our exponential bacterial cultures and centrifuged them for 2 to 3 min at 5,000 rpm. We discarded the supernatant and resuspended the pellet in D–PBS to reach an optical density value of approximately 3. We then loaded 0.5 µl of bacterial culture in the lumen of AirGels (this small volume allowed for ALI maintenance). The resulting multiplicity of infection was approximately 10 (approximately $10^6$ bacteria for approximately $10^5$ airway cells). For the infection shown in Fig 3A, we started with a stationary *P. aeruginosa* culture that we diluted in D–PBS to an optical density of approximately 0.035. We then dipped a sterile toothpick in the culture and lightly touched the inlet of an AirGel with it in order to deposit bacteria while maintaining the ALI. This second method may mechanically compromise the epithelium with the toothpick; we therefore opted for the first one in most experiments.

The chips were then placed in an UNO–T–H–CO2 stage–top incubator (Okolab) for temperature, humidity, and $CO_2$ control. The environment was maintained at 37°C and 5% $CO_2$ and connected to a bottle of Milli–Q water for humidification. Since condensation frequently appears on the PDMS chip during imaging, we placed pieces of Kimtech Science Kimwipes

(Kimberly–Clark Professional) in the inlet ports of AirGels; this prevented dripping water from disrupting the ALI conditions. We visualized the infection progress over time with the aforementioned spinning disk confocal microscope. For WT, *ΔpilT*, and *ΔpilH*, we repeated the infections to reach *N* = 3 replicates per condition. The AirGels for all 3 replicates were all made from the same healthy donor and were between 33– and 38–day–old at the time of infection.

### Quantification of bacterial growth on airway mucus

To compare the growth rates of the WT, *ΔpilT*, and *ΔpilH* strains on mucus, we grew all 3 strains and concentrated them in D–PBS as indicated above to reach a starting optical density of 3. We loaded 0.75 μl of the resulting culture into NHBE cells growing at the ALI on Transwell membranes (Corning) with a 0.4 μm pore size. We infected *N* = 3 wells with each strain. In parallel, we also plated the inoculum to count CFUs. After 5 h incubation of the Transwells in a cell culture incubator (set to 37°C, 5% $CO_2$), we dissociated the infected airway tissue using a 0.1% Triton X–100 (VWR Life Science) solution and mechanical disruption by pipetting and vortexing vigorously. We plated the extracted bacteria for a CFU count on LB–agar and incubated overnight at room temperature. We finally calculated the number of generations $n_{gen}$ as follows:

$$n_{gen} = \log_2\left(\frac{Final\ CFUs}{Initial\ CFUs}\right).$$

### Colonization of extracted mucus

We isolated mucus from 8.5–month–old NHBE cells grown on 0.4 μm pore size polyester Transwell membranes (Corning). To do so, we immersed the apical side of the membranes in a jacalin–fluorescein solution (50 μg/ml in D–PBS) and we placed them in a cell culture incubator for 30 min. We then collected all fluid from the apical side with a pipette and dispensed 12.5 μl into 4 mm PDMS gaskets bonded to a glass–bottom dish (1.5 coverslip, glass diameter 20 mm, MatTek). We filled the space around the PDMS gasket with D–PBS to prevent dehydration of the mucus. We then centrifuged an exponential *P. aeruginosa* mScarlet culture at 5,000 rpm for 3 min and resuspended them in D–PBS before loading 15 μl on the labeled mucus. We then placed the dish in the stage–top incubator and imaged the bacteria and mucus with the confocal spinning disk microscope described above.

### Twitching motility on mucus

We infected a 4–month–old HBE Transwell with *P. aeruginosa* mScarlet as follows. We loaded 3.3 μl of early exponential culture (approximately $10^5$ CFUs) on the apical side of the HBE culture, which had been labeled with fluorescent jacalin. We carefully took the Transwell insert out of the cell culture plate using sterile tweezers and we placed it on a glass–bottom dish (1.5 coverslip, MatTek). We then placed the dish in a stage–top incubator and recorded time–lapse videos of twitching bacteria with our spinning–disk confocal microscope. Because of the lack of culture medium in the visualization setup, recordings could not last long and would dehydrate within minutes.

### Biofilm image acquisition and analysis

We acquired z–stack of infected AirGels over a 35–μm deep range at different time points (*t* = 0 h, 2 h, 4 h, and 5.5 h ± 0.5 h). All the image analysis steps were done in Jupyter Notebooks [72].

Since the AirGel surface is curved, for all subsequent steps, we projected images in 2D using the maximal intensity projection tool in Fiji in order to facilitate downstream analysis. We started by quantifying the sizes of bacterial clusters. First, we visually inspected the pictures: if there were large intensity variations (e.g., in case of a mix of dim single cells and bright clusters), we saturated bright pixels to 1.5 times the mean intensity of the picture. We then segmented the pictures using Otsu thresholding (from the "opencv" Python package [75], version 4.5.4.60) and visually assessed the result. In the rare cases where the segmentation was not deemed satisfactory (i.e., if some features were not detected properly or if there was too much noise), a simple threshold was manually selected. The pictures were then closed and filtered; more specifically, we removed any object smaller than approximately 6 μm$^2$ (20 pixels), which approximately corresponds to the area of a single cell. We then obtained the area of each cluster using the function "regionprops" ("scikit–mage" Python package [76], version 0.19.2), which calculates properties of segmented objects in binary pictures. We calculated the mean cluster area for each replicate; then, for each condition, we plotted the maximum, minimum, and mean of the means (e.g., Fig 3B). We also computed and plotted the proportion of aggregates larger than 100 μm$^2$ (S4 Fig).

We then quantified colocalization between mucus and bacteria. The segmentation and filtration of mucus pictures was identical as for bacterial clusters. Then, using the logical "&" function, we identified the pixels that were common between the binary pictures from the bacterial and mucus channels. With "regionprops," we obtained the areas of these common zones and we normalized them to the total area of mucus. Thus, we could find the proportion of mucus that was covered in bacteria. We finally calculated the proportion of mucus devoid of bacteria as follows: 1 –(proportion of mucus covered in bacteria).

To quantify the contraction of a patch of mucus, we first canceled the effects of drift by registering the images in Fiji using the "Correct 3D drift" plugin. We then manually tracked the displacement of $N = 7$ reference features with the Fiji plugin "Manual Tracking." We loaded the trajectories in a Jupyter Notebook and calculated the distances between each pair of positions over time. We finally normalized the resulting data to the initial distances and plotted them, along with the mean and standard deviation at each time point (Fig 4D).

Finally, we also measured mucus shrinkage over time for WT, *ΔpilH*, and an uninfected control AirGel. To do so, we used images from 30 min timelapses (the starting point of the timelapses differed: 6 h 10 min for WT, 2 h 30 min for *ΔpilH*, and 8 h 5 min for the negative control). We segmented and quantified mucus areas as described above for each time point, and normalized it to the initial area (Fig 6C and S8 Video).

## iSCAT–based quantification of type IV pili retraction frequency

*P. aeruginosa* were grown as previously described [77]. Briefly, an overnight culture was obtained from a single colony and grown in LB at 37˚C with 290 rpm shaking. The overnight culture was diluted 1:500 or 1:1,000 and grown for 2 to 3 h to obtain a mid–exponential phase culture. For surface–grown cells, 100 μl of the mid–exponential phase cell suspension were plated on LB 1% agarose, grown for 3 h at 37˚C and harvested in 500 μl LB by gentle scraping. Cells were diluted to OD600 0.02 to 0.05 prior to visualization. Both liquid–and solid–grown cells were either loaded on 500 μm × 140 μm PDMS microchannels or in 6 mm PDMS gaskets. Cells sticking to the surface were visualized without flow with iSCAT and movies were recorded at 10 fps for either 2 min, 1 min, or 30 s. Raw iSCAT images were processed as described previously [59,77]. Individual movies were manually analyzed using Fiji [66] by counting the total number of TFP in each cell as well as the number of TFP retractions represented by tensed TFP. The residence time of each cell on the coverslip was also recorded. For

each cell, we computed the retraction frequency by dividing the total number of retractions by the residence time of the cell on the coverslip. Finally, we computed a bootstrap median retraction frequency and 95% confidence interval by pooling the data obtained by all 2 to 3 biological replicates. Data analysis was performed using Matlab R2020a (Mathworks).

## Statistical analysis

All statistical tests were run in Python using Jupyter Notebooks [72]. Independent or paired–samples Student *t* tests were performed with Bonferroni correction using the function "add_stat_annotation" from the statannot package [78] (version 0.2.3). One–way ANOVAs were run using the function "f_oneway" in the "stats" module from SciPy [79] (version 1.7.3). When the ANOVA result rejected the null hypothesis, we followed up with a post hoc Tukey test using "stats.multicomp.pairwise_tukeyhsd" from the "statsmodels" module [80].

## Computational model of mucus remodeling by T4P

We refer the reader to our previous work [55,81] for the general theory on the kinematics of the surface and the volume of a 3D soft body, focusing on cell–laden microtissues. Specific considerations on the implementation of this work are introduced in the following formulations.

### 1. Kinematics

Let $V$ be a fixed reference configuration of a continuum body $\mathcal{B}$. We use the notation $\chi: V \rightarrow \mathbb{R}^3$ for the deformation of body $\mathcal{B}$. A motion $\chi$ is the vector field of the mapping $\boldsymbol{x} = \chi(\mathbf{X})$, of a material point in the reference configuration $\mathbf{X} \in V$ to a position in the deformed configuration $\boldsymbol{x} \in v$. The kinematics of a material point are described by the following:

$$\mathbf{u}(\mathbf{X}, t) = \boldsymbol{x}(\mathbf{X}, t) - \mathbf{X}, \tag{S1}$$

where $\mathbf{u}(\mathbf{X}, t)$ is the displacement vector field in the spatial description. The kinematics of an infinitesimal bulk element are described by the following:

$$\mathbf{F}(\mathbf{X}, t) = \frac{\partial \chi(\mathbf{X}, t)}{\partial \mathbf{X}} = \nabla_{\mathbf{X}} \boldsymbol{x}(\mathbf{X}, t) \tag{S2}$$

$$\mathbf{F}^{-1}(\boldsymbol{x}, t) = \frac{\partial \chi^{-1}(\boldsymbol{x}, t)}{\partial \boldsymbol{x}} = \nabla_x \mathbf{X}(\boldsymbol{x}, t), \tag{S3}$$

where $\mathbf{F}(\mathbf{X}, t)$ and $\mathbf{F}^{-1}(\boldsymbol{x}, t)$ are the deformation gradient and inverse deformation gradient, respectively. Note that $J(\mathbf{X}, t) = \mathrm{d}v/\mathrm{d}V = \det \mathbf{F}(\mathbf{X}, t)$ is the Jacobian determinant defining the ratio of a volume element between material and spatial configuration.

A motion of an arbitrary differential vector element can be mapped by the deformation gradient $\mathbf{F}$. However, a unit normal vector $\mathbf{N}$ in the reference configuration cannot be transformed into a unit normal vector $\mathbf{n}$ in the current configuration via the deformation gradient [82], motivating us to develop the kinematics of an infinitesimal surface element [83]. Note that we utilize $\{\hat{\bullet}\}$ to denote the surface quantity bounded by outer surface denoted as $\partial\Omega_0$.

$$\hat{\mathbf{F}}(\mathbf{X}, t) = \frac{\partial \chi(\mathbf{X}, t)}{\partial \mathbf{X}} \cdot \hat{\mathbf{I}} = \hat{\nabla}_{\mathbf{X}} \boldsymbol{x}(\mathbf{X}, t) \tag{S5}$$

$$\hat{\mathbf{F}}^{-1}(\boldsymbol{x}, t) = \frac{\partial \chi^{-1}(\boldsymbol{x}, t)}{\partial \boldsymbol{x}} \cdot \hat{\boldsymbol{i}} = \hat{\nabla}_x \mathbf{X}(\boldsymbol{x}, t), \tag{S6}$$

where $\hat{\mathbf{F}}(\mathbf{X}, t)$ and $\hat{\mathbf{F}}^{-1}(\boldsymbol{x}, t)$ are the deformation gradient and inverse deformation gradient, respectively. Note that $\hat{\mathbf{I}} = \mathbf{I} - \mathbf{N} \bigotimes \mathbf{N}$ and $\hat{\boldsymbol{i}} = \boldsymbol{i} - \mathbf{n} \bigotimes \mathbf{n}$ are the mixed surface unit tensors, where $\mathbf{I}$ and $\boldsymbol{i}$ are the unit tensors, and $\mathbf{N}$ and $\mathbf{n}$ are the outward unit normal vectors, in reference and current configuration, respectively. Note that $\hat{J}(\mathbf{X}, t) = \mathrm{d}a/\mathrm{d}A = |\mathrm{cof}\ \mathbf{F} \cdot \mathbf{N}|$ is the Jacobian determinant defining the ratio of an area element between material and spatial configuration.

## 2. Equilibrium

The total potential energy functional $W(\chi)$ is defined as:

$$W(\chi) = \int_{\Omega_0} \Psi(\mathbf{F}, \chi; \mathbf{X})\mathrm{d}V + \int_{\partial\Omega_0} \hat{\Psi}(\hat{\mathbf{F}}, \chi; \mathbf{X})\mathrm{d}S - \int_{\Omega_0} \mathbf{B} \cdot \mathbf{u}(\chi; \mathbf{X})\mathrm{d}V - \int_{\partial\Omega_0} \mathbf{T} \cdot \mathbf{u}(\chi; \mathbf{X})\mathrm{d}S, \quad \text{(S9)}$$

where $\Psi$ and $\hat{\Psi}$ are strain energies in bulk and on surface, and $\mathbf{B}$ is the reference body force and $\mathbf{T}$ is the surface traction. An equilibrated configuration is obtained by minimizing this functional considering all admissible deformations $\delta\chi$. It is important to note that the strain energies ($\Psi, \hat{\Psi}$) can be varied depending on the bacterium and mucus models so that the following sections can be written in a single formulation for brevity, and the specific strain energies are to be defined in section 4.

Following the derivation presented in [84], we can finally arrive at a set of localized force balance equations. Neglecting the inertial effect, the local form of linear and angular momentum balances for bulk and surface are defined by the following:

$$\nabla_{\mathbf{X}} \cdot \mathbf{P} + \mathbf{B} = \mathbf{0} \text{ in } V \tag{S10}$$

$$\hat{\nabla}_{\mathbf{X}} \cdot \hat{\mathbf{P}} + \mathbf{T} - \mathbf{PN} = \mathbf{0} \text{ on } S \tag{S11}$$

$$\mathbf{u} = \check{\mathbf{u}} \text{ on } S_u \tag{S12}$$

$$[\![\hat{\mathbf{P}}\hat{\mathbf{N}}]\!] = 0 \text{ on } L,$$

where $\check{\mathbf{u}}$ is the prescribed displacement on the boundary $S_u$, $\hat{\mathbf{N}}$ is the bi–normal vector to the boundary curve, and $[\![\bullet]\!]$ indicates summation over surfaces intersecting on boundary curves [83].

## 3. Weak form

For the finite element implementation, we need to obtain the weak form for our problem. By adding the constraint that the first variation of the total potential energy must be equal to zero $\delta W(\chi) = 0$, we obtain a weak form statement as

$$G = \int_{\Omega_0} \mathbf{P} : \nabla_{\mathbf{X}}\delta\mathbf{u}\ \mathrm{d}V + \int_{\partial\Omega_0} \hat{\mathbf{P}} : \hat{\nabla}_{\mathbf{X}}\delta\mathbf{u}\ \mathrm{d}S - \int_{\Omega_0} \mathbf{B} \cdot \delta\mathbf{u}\ \mathrm{d}V - \int_{\partial\Omega_0} \mathbf{T} \cdot \delta\mathbf{u}\ \mathrm{d}S = 0 \forall \delta\mathbf{u}, \quad \text{(S13)}$$

where $\delta\mathbf{u}$ is the admissible deformation field.

We employed the open–source platform FEniCS [85], to implement the finite element simulation. We used the Scalable Nonlinear Equations Solvers (SNES) from the open–source toolkit PETSc [86], which provides numerical computations of a Newton–type iterative procedure to solve the nonlinear variational problem. Note that the value of $\gamma$ should be ramped from zero to its prescribed value for numerical stability as the problem is highly nonlinear.

## 4. Constitutive relations

To relate the active stress with deformation, we must specialize our choice for the strain energies in the bulk and on the surface. For the deformation of compressible tissue, we consider a passive bulk energy $\Psi^p$ that captures the permanent elasticity of the collagen network, and for the contribution of bacterium contractility, we can consider the active surface energy $\hat{\Psi}^a$ that accounts for the action of the bacterium on the surface of mucus tissue.

**4.1. Passive mucus model.** The passive strain energy density $\Psi^p$ describes the elasticity of mucus tissue. We consider the mucus as a soft, highly deformable, and highly compressible hyperelastic material, but we neglect its biphasic and viscoelastic nature in terms of energy dissipation. We choose the compressible neo–Hookean model [81,82] for the mucus.

$$\Psi^p = \frac{K}{2}(J-1)^2 + \frac{G}{2}(I_1 - 3 - 2\ln J),$$ (S14)

where $K$ and $G$ are the bulk and shear moduli.

**4.2. Active model for the contractile action of bacteria.** We assume that the bacteria–mucus interaction can be described through a surface strain energy generating constant surface stresses similar to fluid–like surface tension [55,81]. Bacteria exert a contractile force on the periphery of the mucus, and we recapitulate this action through an active surface energy $\hat{\Psi}^a$. We postulate that the surface energy $\hat{\Psi}^a$ is a function of the change of the surface area $\hat{J}$.

$$\hat{\Psi}^a = \gamma \hat{J},$$ (S15)

where $\gamma$ is a surface contractile modulus (energy per unit area) representing the contractility of bacterium on the surface at the equilibrium state. It is important note that we consider no bulk contractility due to the bacteria as results verify their presence only on the periphery of the mucus.

**4.3. Energy penalization.** As at high level of contraction the bacteria are bound to jam, barring the additional contraction of the mucus (even if the material itself can accommodate it), we have to enforce this jamming transition. Assuming that we know the initial surface concentration of bacteria, we enforce the kinematic constraint via energy penalization. From the experimental observation, we enforce the surface area of deformed mucus tissue cannot be smaller than a ratio ($\hat{J}_{pen}$) of the initial surface area. An appropriate energy penalization $\hat{\Psi}_{pen}$ for enforcing the prescribed surface condition is given by

$$\hat{\Psi}_{pen} = \frac{P}{2}\langle \hat{J}_{pen} - \hat{J} \rangle^2,$$ (S16)

where $P$ is the penalty parameter (energy per unit area), and $\langle \bullet \rangle$ is the Macaulay brackets that used to describe the ramp function.

$$\langle x \rangle = \begin{cases} x \ (x > 0) \\ 0 \ (x \leq 0) \end{cases}$$ (S17)

## 5. Finite element simulation

The reference (undeformed) state corresponds to a state where the active contractile moduli are set to zero. Experimentally, this reference state corresponds to the initial state of the mucus right after the mixing of mucus and bacterium and before the application of forces by encapsulated bacterium. The reference configuration for the finite element simulations represents the geometry shown in Fig 5B. The entire surface is allowed to actively contract through increasing the surface contractile modulus up to an equilibrium value. The final (deformed) state is

defined when the surface contractile moduli $\gamma$ reaches its prescribed value, and no external loads are applied. Experimentally, this corresponds to the equilibrium state of the mucus. The final configurations represent the equilibrium states.

## 6. Parameter calibration

The parameters of the model are the bulk and shear moduli, $K$ and $G$, and surface contractile modulus, $\gamma$, penalty parameter, $P$ and penalty surface ratio, $\hat{J}_{pen}$. There is a unique relationship between $K$, $G$, and Poisson's ratio $\nu$, allowing to interchangeably use $\nu$ in place of $K$ for the calibration procedure. We selected a set of parameters: $G = 1.0$ Pa, $\nu = 0.1$, $\gamma = 0.03$ nN/μm and $P = 1.0$ nN/μm. The corresponding bulk and elastic moduli were $K = 0.9$ Pa and $E = 2.2$ Pa within the range of reported values for mucus [87,88].

## Supporting information

**S1 Fig. Gene expression profiles of AirGels measured by single cell RNA–seq.** (**A**) The expression of marker genes of lung epithelial cell types is shown for each cluster defined from the scRNA–seq reads of AirGels. Average expression per cluster and the percentage of cells expressing the respective marker gene. (**B**) Subset–specific expression of canonical marker genes on UMAP embedding. FOXJ1 and PIFO are typically expressed in ciliated cells. The immature ciliated cell cluster, also known as deuterosomal cells, is marked by high levels of FOXJ1 and expression of FOXN4. Basal cells typically express TP63 and KRT5. The secretory cluster shows expression of SCGB1A1 and a fraction of more mature secretory cells expressing MUC5B [89]. Furthermore, we observe a transitional state between basal and secretory, the immature secretory cluster, which shows partial mutual expression of KRT4 and KRT13 as previously described [40]. (**C**) A gene ontology (GO) analysis was performed on the most differentially expressed genes in each cluster confirming the correct annotation of cell clusters. (PNG)

**S2 Fig. AirGels produce MUC5AC and MUC5B simultaneously.** Luminal mucus in a methacarn–fixed AirGel. Staining was done with antibodies against MUC5AC and MUC5B gel–forming mucins. (PNG)

**S3 Fig.** (A) Velocities of fluorescent microparticles transported by the cilia of healthy airway epithelia growing on Transwell (TW) membranes. (B) Maximal (red triangles) and median (gray circles) velocities of fluorescent microparticles for N = 3 Transwells, differentiated AirGels, and undifferentiated AirGels, respectively. The low median values for differentiated AirGels can be at least partly explained by the curvature of the lumen: when we image AirGels with a confocal microscope, we simultaneously visualize particles at varying distances of the epithelium. Only a fraction of these particles lies within the distance that allows for maximal clearance, thereby biasing population–level velocities to lower values. (PNG)

**S4 Fig. Distribution of cluster areas at $t = 0$ h and $t = 5.5$ h during 1 infection experiment.** The red line indicates the threshold for what we considered as large clusters ($>100$ μm$^2$). (PNG)

**S5 Fig. Infection of an AirGel in which mucus was not labeled.** After 4 h 15, bacterial aggregates were already visible, indicating their formation is independent of jacalin staining. (PNG)

**S6 Fig. *P. aeruginosa* does not form biofilms in AirGels lacking mucus, but damages its tissue more rapidly.** Orthogonal views of infections with *P. aeruginosa* in a differentiated versus an undifferentiated AirGel stained with Jacalin at low **(A)** and high **(B)** magnification. The infection was imaged at $t = 5.5$ h (differentiated) and $t = 6$ h (undifferentiated) post–inoculation. L indicates the luminal side and M the extracellular matrix. We did not observe *P. aeruginosa* aggregates on the luminal side of the undifferentiated AirGel. However, bacteria damaged the epithelium extensively in the absence of mucus, which resulted in invasion of the extracellular matrix (white arrowheads). Magenta: plasma membrane; green: mucus; orange: *P. aeruginosa*.
(PNG)

**S7 Fig. *P. aeruginosa* does not form biofilms on mucus extracted from HBE cultures.** Jacalin–labeled mucus that had been isolated from a differentiated HBE culture on a Transwell. Even after almost 6 h after incubation with *P. aeruginosa*, no aggregates were visible.
(PNG)

**S8 Fig. Mucus does not contract in the absence of *P. aeruginosa*.** Jacalin–stained mucus in an uninfected AirGel. The total area of mucus was estimated over time and found to only differ slightly over time, most likely due to photobleaching and drift in and out of focus.
(PNG)

**S9 Fig. Infection of an AirGel with AP1913, a clinical *P. aeruginosa* isolate.** This strain came from the bronchoalveolar lavage fluid of a cystic fibrosis patient. Micrographs show a time course during which bacteria (orange) contract mucus (green). All images are maximal intensity projections from z–stacks.
(PNG)

**S10 Fig. T4P are necessary for mucus contraction but not for attachment.** Aggregates of the *ΔpilA* mutant *P. aeruginosa* were still absent 3 h 25 post–inoculation. However, the bacteria colocalized with the mucus, indicating that T4P are not necessary for adhesion to mucus.
(PNG)

**S11 Fig. T4P retraction frequency increases during surface contact and is constitutively high in a *pilH* deletion mutant.** T4P retraction rates were measured by interferometric scattering (iSCAT) microscopy, which allows for label–free T4P visualization [59]. To prevent cells from swimming away during the iSCAT measurements, a flagellum–less *ΔfliC* mutant was used as background strain. This strain was either grown in liquid or preadapted to culture on a solid surface for 3 h. Black circles and bars indicate the bootstrap median and 95% confidence interval of the medians, respectively.
(PNG)

**S1 Video. Real–time video recording of beating cilia in a differentiated AirGel.**
(MP4)

**S2 Video. Network of mucus–associated *P. aeruginosa* biofilms 13 h post–inoculation into an AirGel.** Beating of the underlying ciliated epithelium shakes the biofilms, but is not sufficient to clear them out of the model airway.
(AVI)

**S3 Video. Time–lapse spinning disk confocal micrographs showing *P. aeruginosa* (orange) on jacalin–labeled mucus (green) isolated from a Transwell.** Even 5 h post–infection, the bacteria did not aggregate on nor remodel the mucus.
(AVI)

**S4 Video. Time–lapse spinning disk confocal micrographs of *P. aeruginosa* (orange) twitching on the mucus (green) of an HBE culture in a Transwell insert.** The bacteria were loaded 3.5 h before the time–lapse was recorded.
(AVI)

**S5 Video. Time–lapse spinning disk confocal micrographs of *P. aeruginosa* (orange) on jacalin–labeled mucus (green).** The initial time stamp in the video corresponds to *t* = 6 h 10 post–inoculation. Several clusters can be seen fusing over the course of the experiment; large pieces of mucus also shrink upon interactions with bacteria.
(AVI)

**S6 Video. Time–lapse spinning disk confocal micrographs of mucus contraction by AP1913, a *P. aeruginosa* clinical strain.** It was isolated from the bronchoalveolar lavage fluid of a cystic fibrosis patient and used to infect an AirGel. Mucus is shown in green, *P. aeruginosa* in orange.
(AVI)

**S7 Video. Time–lapse spinning disk confocal micrographs of a T4P–less *P. aeruginosa* *ΔpilA* (orange).** The bacteria are moving along with the flowing, jacalin–labeled mucus (green), indicating that T4P are not necessary for adhesion to mucus.
(AVI)

**S8 Video. Compilation of 30 min time–lapses showing jacalin–labeled mucus (green) in different conditions: first, without any bacteria, then exposed to a WT *P. aeruginosa* strain (orange) and finally exposed to a hyperpiliated *ΔpilH* mutant.** Since *ΔpilH* starts aggregating and remodeling mucus earlier than WT, the starting points of the recording differed (*ΔpilH*: 2 h 30 min, WT: 6 h 10 min, negative control: 8 h 5 min).
(AVI)

**S9 Video. Time–lapse spinning disk confocal micrographs of the hyperpiliated *P. aeruginosa* *ΔpilH* mutant (orange) rearranging jacalin–labeled mucus (green).** Dramatic remodeling of the mucus starts as early as 1.5 h post–inoculation, resulting in numerous mucus–associated aggregates.
(AVI)

**S1 Data. scRNA–seq data underlying the plot shown in Fig 1F.**
(XLSX)

**S2 Data. CBF and clearance velocity data used to generate the plots from Fig 2B and 2C.**
(XLSX)

**S3 Data. Raw data of cluster areas and colocalization between mucus and bacteria, used to generate the plots from Fig 3B and 3C.**
(XLSX)

**S4 Data. Distances between reference features in mucus over time, underlying the graph in Fig 4D.**
(XLSX)

**S5 Data. Various datasets underlying the plots shown in Fig 5B, 5C and 5D.**
(XLSX)

**S6 Data. Cluster areas and mucus contraction data used to generate Fig 6B and 6C.**
(XLSX)

**S7 Data. Gene expression data underlying S1A and S1B Fig.**
(ZIP)

**S8 Data. Data from gene ontology analysis underlying S1C Fig.**
(XLSX)

**S9 Data. Clearance velocity data used to generate the plots shown in S3 Fig.**
(XLSX)

**S10 Data. Cluster area dataset used to generate S4 Fig.**
(XLSX)

**S11 Data. T4P retraction data underlying S11 Fig.**
(XLSX)

**S1 Table. *P. aeruginosa* strains used in this study.**
(DOCX)

**S2 Table. Plasmids used in this study.**
(DOCX)

**S3 Table. Growth of different *P. aeruginosa* mutants on airway mucus.**
(DOCX)

## Acknowledgments

We thank Zaïnebe Al–Mayyah, Jeremy Wong, the Bioimaging and Optics Core Facility (BIOP), and the Gene Expression Core Facility (GECF) at EPFL for technical assistance. Dr. Marco Kühn, Dr. Johannes Bues, and Dr. Sophia Hsin–Jung Li for insightful discussions; Dr. Chantal Quiblier, Dr. Tim Roloff, and Prof. Dr. Adrian Egli and Dr. Romé Voulhoux for strains and Formlabs forum user Telliria for suggestions on 3D–printing process.

## Author Contributions

**Conceptualization:** Tamara Rossy, Alexandre Persat.

**Formal analysis:** Tamara Rossy, Tania Distler, Joern Pezoldt, Jaemin Kim, Lorenzo Talà.

**Funding acquisition:** Alexandre Persat.

**Investigation:** Tamara Rossy, Tania Distler, Lucas A. Meirelles, Lorenzo Talà.

**Methodology:** Tamara Rossy, Tania Distler, Jaemin Kim, Nikolaos Bouklas, Alexandre Persat.

**Project administration:** Tamara Rossy, Alexandre Persat.

**Software:** Tamara Rossy, Joern Pezoldt, Jaemin Kim.

**Supervision:** Alexandre Persat.

**Visualization:** Tamara Rossy, Tania Distler.

**Writing – original draft:** Tamara Rossy, Alexandre Persat.

**Writing – review & editing:** Tamara Rossy, Tania Distler, Lucas A. Meirelles, Joern Pezoldt, Jaemin Kim, Nikolaos Bouklas, Bart Deplancke, Alexandre Persat.

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
