## [Editor Report · Decision Letter 0]

15 Dec 2022

Dear Dr. Persat, 

Thank you for submitting your manuscript entitled "Pseudomonas aeruginosa contracts mucus to form biofilms in tissue-engineered human airways" for consideration as a Research Article by PLOS Biology.

Your manuscript has now been evaluated by the PLOS Biology editorial staff, as well as by an academic editor with relevant expertise, and I am writing to let you know that we would like to send your submission out for external peer review.

Once your full submission is complete, your paper will undergo a series of checks in preparation for peer review. After your manuscript has passed the checks it will be sent out for review. To provide the metadata for your submission, please Login to Editorial Manager (https://www.editorialmanager.com/pbiology) within two working days, i.e. by Dec 17 2022 11:59PM.

Kind regards,

Paula

---

Senior Editor

PLOS Biology

---

## [Decision Letter · Decision Letter 1]

6 Mar 2023

Dear Dr. Persat,

Please allow me to first apologize for the delay in the processing of your manuscript. This delay is caused by my difficulty in recruiting reviewers for your manuscript, and is further compounded by one referee promising an overdue report but failing to deliver after long delay and multiple chases. I am sorry for this unexpected event, and I thank you for your patience while your manuscript "Pseudomonas aeruginosa contracts mucus to form biofilms in tissue-engineered human airways" was peer-reviewed at PLOS Biology. It has now been evaluated by the PLOS Biology editors, an Academic Editor with relevant expertise, and by several independent reviewers. 

In light of the reviews, which you will find at the end of this email, we would like to invite you to revise the work to thoroughly address the reviewers' reports.

As you will see below, the reviewers find the manuscript interesting but raise some issues that should be solved before publication. Please address all the reviewers' issues including the control experiments with the natural isolate of P. aeruginosa from the CF patient, as requested by the reviewer #2. 

Given the extent of revision needed, we cannot make a decision about publication until we have seen the revised manuscript and your response to the reviewers' comments. Your revised manuscript is likely to be sent for further evaluation by all or a subset of the reviewers.

**IMPORTANT - SUBMITTING YOUR REVISION**

*Re-submission Checklist*

*Published Peer Review*

*PLOS Data Policy*

*Blot and Gel Data Policy*

Sincerely,

Paula

---

Senior Editor

PLOS Biology

REVIEWS:

Reviewer #1: Biophysics.

Reviewer #2: Airway mucus, cystic fibrosis and Pseudomonas aeruginosa.

Reviewer #1: Rossy et al establish a novel lung model system for studying Pseudomonas aeruginosa infection. This model system enables 3D live imaging of bacteria. The authors confirm that after differentiation, epithelia are protected from damage caused by P. aeruginosa as expected for mucus producing epithelia. They show that within this model system, P. aeruginosa forms biofilms at a time scale of hours, mostly driven by fusion of colonies. Importantly, the time scale of biofilm formation is much faster than has been observed in vitro so far. The authors identify type IV pilus activity as the mechanism for mucus contraction and colony fusion. They conclude that mucus formation comes with a trade-off; while it protects the epithelia from bacterial invasion, it facilitates biofilm formation.

I am no expert on infection models, but the model developed in this study looks intriguing, since it reveals bacterial behavior that is not reproducible in vitro. Overall, I believe the authors established a highly promising infection model that will serve the community to address follow up questions about biofilm formation by an important model organism. The result that type IV activity is required for rapid biofilm formation in the close to natural infection environment are both convincing and interesting. I have only a few minor issues that hopefully help improving the manuscript.

1. While the model shows previously unseen behavior, it remains unclear whether mucus contraction affects pathogenicity of P. aeruginosa. It would be good to discuss this point.

2. Line 230: "After this initial biofilm seeding, P. aeruginosa may initiate the secretion of matrix components to strengthen the cohesion of the biofilm." On which experimental hint or evidence is this speculation based? Since this speculation is not required for further discussion I suggest omitting it.

3. The dynamics of colony formation and fusion are astonishingly similar to colony formation of Neisseria species in vitro even in the absence of mucus. It would be interesting to discuss this analogy. (doi.org/10.1016/j.cell.2018.04.010 and doi.org/10.1103/PhysRevLett.121.118102)

4. There is no reference to Supplementary Fig. 2a. Why is its shown? Is it part of the control for jacalin labeling?

5. Supplementary Fig. 3: It would be good to show the fluorescence signals of bacteria (orange) and membrane? (pink) in different panels. It is difficult to distinguish these colors. Also, please explain what the colors refer to the Figure legend.

6. Supplementary Fig. 8: I understand that the total fluorescence intensity is used as a measure for total cell number. If this is correct then the there seems to be no net growth. How can you claim that the growth rate is not influenced by mutations? What means N = 9? There are only three curves per strain. Please correct to N = 3 if there are only three curves per strain.

Reviewer #2: This paper connects important airway biology and microbiology processes in a way that is novel, fills a gap in the field, and addresses previously unknown mechanisms of bacterial function and disease. There are, however, some gaps in the description of the methods and in the results that should be addressed before publication.

1. Figure 1 presents an interesting and innovative method of culturing airway epithelial cells in a more physiologically relevant way. However, it's not clear how the cells are able to maintain air-liquid interface. The diagram in Figure 1b shows a luminal access port, but how is this port protected from the surrounding media and how is air moved in? Is there a pressure pump moving air across? Perhaps a picture of the actual device rather than a graphic would make the ALI more clear.

2. The methods section indicates that HBEs were obtained from both CF and non-CF sources. Which were used in the data presented here?

3. Figure 2 nicely shows appropriate architecture. However, MUC5AC and MUC5B should be differentially stained, to understand which mucin is primarily expressed. This is particularly important for the later figures, because previous studies have shown that the arrangement of mucins in the airway strands may influence disease.

4. In figure 2 iii, the measured mucociliary clearance is very slow. This is important because the rate of mucus clearance could affect biofilm formation. The references cited to defend this rate as physiologic are a very small subset of many studies measuring MCC ex vivo, (one of the papers cited is a review, which does not present primary data) and one of the papers present MCC data that is ten-fold higher than the data presented here. The authors should include more references to capture a more complete picture of the data found in the field, and a comparison of MCC and ciliary beat frequency measured by these methods in flat monolayers, which have been shown to closely represent ciliary function, would be needed to ensure that the method of culturing is not affecting mucus and cilia movement. This is particularly important given that these host factors may influence mucus contraction in addition to the bacterially-driven mechanisms proposed later in the study.

5. There is some important information missing from the methods regarding the infections. At what CFU are the bacteria inoculated? And at what stage of growth? The formation of biofilms in the mucus may be dependent on if the bacteria are collected in the doubling or stationary phases.

6. The study as presented is convincing, particularly around with the mutants. However, confirmatory studies showing this relationship with mucus in a non-PAO1 strain would be a critical component. Of particular interest would be a lung-adapted strain. This is vital, because patients, especially with CF and other muco-obstructive disorders, have cultured isolates of P. aeruginosa that are phenotypically and mechanistically very different from PAO1.

---

## [Editor Report · Decision Letter 2]

6 Jun 2023

Dear Dr Persat,

Thank you for your patience while we considered your revised manuscript "Pseudomonas aeruginosa contracts mucus to form biofilms in tissue-engineered human airways" for publication as a Research Article at PLOS Biology. This revised version of your manuscript has been evaluated by the PLOS Biology editors and by the Academic Editor. I'm handling your paper temporarily while my colleague Dr Paula Jauregui is out of the office.

Based on our Academic Editor's assessment of your revision, we are likely to accept this manuscript for publication, provided you satisfactorily address the following data and other policy-related requests.

a) The unusual transitive use of "contracts" in the Title may cause confusion. Please change your Title to "Pseudomonas aeruginosa type IV pili actively induce mucus contraction to form biofilms in tissue-engineered human airways"

b) Please address my Data Policy requests below; specifically, we need you to supply the numerical values underlying Figs 2AB, 3, 4BC, S1, S2, S3, S5ABCD, S6AB, either as a supplementary data file or as a permanent DOI’d deposition.

c) Please cite the location of the data clearly in all relevant main and supplementary Figure legends, e.g. “The data underlying this Figure can be found in S1 Data” or “The data underlying this Figure can be found in https://doi.org/10.5281/zenodo.XXXXX”

d) We note that your Materials and Methods section is currently in the Supplement – please move it to the main manuscript.

We expect to receive your revised manuscript within two weeks. 

*Published Peer Review History*

*Press*

Sincerely,

Roli Roberts

Roland G Roberts PhD

Senior Editor

rroberts@plos.org

PLOS Biology

on behalf of

Editor,

pjaureguionieva@plos.org,

PLOS Biology

DATA POLICY:

Regardless of the method selected, please ensure that you provide the individual numerical values that underlie the summary data displayed in the following figure panels as they are essential for readers to assess your analysis and to reproduce it: Figs 2AB, 3, 4BC, S1, S2, S3, S5ABCD, S6AB. NOTE: the numerical data provided should include all replicates AND the way in which the plotted mean and errors were derived (it should not present only the mean/average values).

DATA NOT SHOWN?

---

## [Editor Report · Decision Letter 3]

21 Jun 2023

Dear Dr Persat,

Thank you for the submission of your revised Research Article "Pseudomonas aeruginosa type IV pili actively induce mucus contraction to form biofilms in tissue-engineered human airways" for publication in PLOS Biology. On behalf of my colleagues and the Academic Editor, Victor Sourjik, I am pleased to say that we can in principle accept your manuscript for publication, provided you address any remaining formatting and reporting issues. These will be detailed in an email you should receive within 2-3 business days from our colleagues in the journal operations team; no action is required from you until then. Please note that we will not be able to formally accept your manuscript and schedule it for publication until you have completed any requested changes.

PRESS

Sincerely, 

Paula

---

Senior Editor

PLOS Biology
